# Economic Growth Target, Government Expenditure Behavior, and Cities' Ecological Efficiency—Evidence from 284 Cities in China

Can Zhang [1] , Tengfei Liu [2], Jixia Li [1], Mengzhi Xu [1], Xu Li [3] and Huachun Wang [1,*]

1 School of Government, Beijing Normal University, Beijing 100875, China
2 School of Business Administration, The Open University of China, Beijing 100039, China
3 China Life Reinsurance Company Ltd., Beijing 100033, China
* Correspondence: huachunwang@bnu.edu.cn

**Abstract:** As a composite indicator that incorporates economic efficiency and environmental protection, ecological efficiency is a valuable tool for measuring regional green development and accelerating regional green transformation. As the economy transitions, Chinese economic growth targets affect local governments' behaviors, thereby impacting ecological efficiency. In this study, the ecological efficiency level of 284 cities in China was measured using the EBM-DEA method from 2007 to 2019, and the spatial exploration analysis method and the dynamic double fixed effect spatial Durbin model were applied to analyze urban ecological efficiency's spatial correlations, impacts, and mechanisms. The conclusions are as follows: China's urban ecological efficiency has increased over time. At the spatial level, it shows the distribution characteristics of east > northeast > middle > west. In terms of spatial agglomeration, there are typically spatial agglomerations, high–high agglomerations, and low–low agglomerations in Chinese cities' ecological efficiency. There is an inverted U-shaped relationship between economic growth target and ecological efficiency. According to regional differences, the economic growth target in the eastern region has a U-shaped impact on ecological efficiency, while in the central, northeast, and western cities they have an inverted U-shaped effect on ecological efficiency. In terms of the impact mechanism, through the intermediary effect test, it is found that appropriate economic growth target setting can promote the proportion of energy conservation and environmental protection expenditure and fiscal science and technology expenditure. Excessive economic growth target setting can inhibit the proportion of energy conservation and environmental protection expenditure and fiscal science and technology expenditure. The proportion of energy conservation and environmental protection expenditure and fiscal science and technology expenditure can promote ecological efficiency. The enlightenment is as follows: China should weaken the economic growth target in official promotion assessment, set differentiated economic growth targets for different regions, and increase the proportion of energy conservation and environmental protection expenditure and fiscal science and technology expenditure to promote ecological efficiency.

**Keywords:** economic growth target; proportion of energy conservation and environmental protection expenditure; proportion of fiscal science and technology expenditure; ecological efficiency; dynamic spatial Durbin model





## 1. Introduction

The concept of ecological efficiency was first proposed by Schaltegger in 1990. In general, the idea is to maximize economic output with the least amount of resource consumption and environmental impact [1]. In addition to reflecting the economic achievements and environmental impacts of human activities, it can also reflect the coordination level between economic development and the protection of the ecological environment [2]. Nowadays, ecological efficiency has become an important indicator for measuring urban sustainable development. China's ecological efficiency level is not high, mainly reflected in

the low level of economic development and high level of environmental pollution. In terms of economic development, China ranked 60th in the world in 2021 with a per capita GDP of USD 12,359. At the level of environmental pollution in 2021, 121 Chinese cities exceeded the ambient air quality standard due to pollution [3].

The existing research mainly explored the path to improve ecological efficiency from the aspects of industrial structure, resource allocation, environmental regulation, etc., ignoring the impact of economic growth targets. Most of the economies in transition are focused on economic growth. Recently, China has become the world's fastest growing economy. By setting economic growth targets, China has made great economic progress since reform and opening up [4]. Economic growth target setting not only affects economic growth but also affects environmental quality. However, rapid economic growth comes at a high environmental cost [5]. Since the reform and opening up in 1978, economic growth became the hard indicator for promotion of officials at all levels. Although the government performance assessment is increasingly diversified, the economic growth is the easiest to measure, so it is still the most important assessment indicator [6]. From the 12th National People's Congress of the CPC to the 18th National People's Congress, the goal of doubling economic growth was clearly put forward, and the goal of economic growth became the performance standard that governments at all levels publicly promised [7]. In the process of transforming from a planned economy to a market economy, China has gradually formed a vertical target management system along the path of socialism with characteristics [8]. Under China's political pyramid, the central government holds the power of personnel, and local governments have the ability to intervene in economic development. Officials with high pressure on political promotion will interfere with the economy and operation of enterprises. The central government leads China's economic growth through economic growth goals, which are broken down to governments at all levels through administrative levels. The economic growth target is not only the assessment of the superior government to the subordinate government but also the commitment of the subordinate government to the superior government. China's fiscal decentralization system gives officials full control over resources and administrative decision-making power, enabling local officials to have a strong ability to intervene and dominate economic development [9]. Under this institutional background, if local officials want to achieve performance appraisal standards and career promotion, they must use various resources to promote economic growth.

As a comprehensive indicator, ecological efficiency can measure the level of economic development and the coordination of environmental quality. The improvement of ecological efficiency is of great practical significance for China to accelerate its sustainable development. The research questions of this article are: What is the evolution trend of cities' ecological efficiency in China in terms of time and space? As one of the target management methods of the Chinese government, will economic growth target setting affect ecological efficiency? How does economic growth target setting affect ecological efficiency? The significance of this study is: the new measurement method is used to measure China's urban ecological efficiency, which improves the accuracy of ecological efficiency measurement, exploring the impact and mechanism of economic growth target setting on ecological efficiency, expanding the relevant research on the impact of economic growth target setting, and providing new paths for improving ecological efficiency. The structure of this paper is as follows: the second part is the literature review; the third part is the research hypothesis; the fourth part is the research methods and data sources; the fifth part is the empirical results analysis; the sixth part is the discussion and conclusion.

## 2. Literature Review

In recent years, the literature on economic growth targets has gradually increased. First of all, scholars have carried out a series of studies on economic growth targets and economic growth, environmental pollution, public expenditure, and innovation. In terms of economic development, most studies believed that economic growth target setting can promote economic development, but it also had a negative impact. Setting an economic growth

target increased the area of land transfer and the proportion of fixed asset investment and rapidly expanded output [10]. Although it can ensure the predictability of economic output, the planning system brought about vicious performance competition, overcapacity, and inhibited future economic growth [11]. At the same time, when local governments faced higher pressure of economic growth, the marginal rate of return on the capital of local enterprises declined [12]. In terms of environmental pollution, there is no consensus on the impact of economic growth targets on ecological efficiency in existing studies, which can be roughly divided into aggravating and inhibiting effects and linear and nonlinear effects. When environmental performance was included in the cadre evaluation system, under the political incentive of local officials, economic growth and environmental protection changed from the original substitute relationship to a complementary relationship, and the setting of economic growth targets improved the ecological environment [13]. The economic growth target could reduce carbon emissions by curbing fiscal decentralization and promoting environmental decentralization [14]. However, according to Zhong et al., the government's economic growth target would lead to decreased environmental supervision, fewer investments in environmental protection, and greater pollution of the environment [15]. Air pollution was inversely related to economic growth targets according to Chai et al. Some scholars thought that economic growth targets had an obvious threshold effect on air pollution. The inhibition effect increases with the adjustment of human capital and industrial structure, while the increase of foreign investment aggravated the impact of economic growth targets on air pollution [16].

Government expenditure can effectively affect ecological efficiency. With regard to government fiscal expenditure preferences, the first generation of fiscal decentralization theory believes that residents "vote with their feet" and flow into areas that can provide them with satisfactory public goods [17]. The government has incentives to optimize public services within its jurisdiction, so as to achieve effective allocation of resources. The second generation of fiscal decentralization theory, from the perspective of principal–agent, believes that local governments, as rational people, have incentives for their own interests that are contrary to the public interest [18], which is more in line with China's reality. According to most studies, a high economic growth target would lead to distortions of local government expenditure structures. The government was under the pressure of stable growth when the market-driven growth rate fell below the economic growth target, which encouraged local governments to promote economic growth. The economic growth target can only be achieved through expenditures on economic affairs rather than public services, health, and education [19]. Local governments typically chose infrastructure construction or other methods that can achieve economic growth targets in a short period of time to achieve a challenging economic growth target, which caused improper allocation of resources and slashed investments in science and technology. Meanwhile, the governments emphasized that investing in capital-intensive and labor-intensive industries impeded the development of knowledge-intensive industries [20]. In terms of innovation, the relationship between economic growth target setting and innovation has not yet been agreed upon by scholars, which can be roughly split into promotion theory and inhibition theory. For example, Liu et al. believed that the economic growth target can effectively improve the green urban land-use efficiency [21]. According to Li et al., local government economic growth targets constrained enterprises' technological innovation [22]. Economic growth targets hindered the improvement of green innovation efficiency through fiscal expenditures and market segmentation [23]. Expenditures directly related to the improvement of ecological efficiency are, respectively, fiscal science and technology expenditure and energy conservation and environmental protection expenditure. Energy conservation and environmental protection expenditure will certainly fill the space for local government environmental governance, enrich the scope and means of governance, and improve environmental quality [24]. However, it will also squeeze funds for production and economic growth. Fiscal expenditure on science and technology mainly refers to the fiscal revenue used to achieve the targets of scientific development, technological transformation, and innovation, reflecting the support

of the central or local governments for science and technology activities [25]. It can provide strong support for scientific and technological innovation activities, so as to improve the level of ecological efficiency. Energy conservation and environmental protection expenditure has the effect of pollution control, and as an input type of environmental regulation it can guide society to increase environmental protection efforts, which is conducive to environmental protection and ecological efficiency, and improve the public service level of the jurisdiction. The above public services can not only improve people's livelihood and welfare but also improve regional ecological efficiency. Government financial expenditure on science and technology can guide and promote industrial development, promote enterprise technological progress, and improve total factor productivity and has leverage and spillover effects on enterprise innovation [18]. Due to the mutual imitation and benchmarking competition strategy between local governments, China's fiscal expenditure on science and technology has shown an upward trend [26]. As barriers between regions are gradually broken, capital, labor, and other factors flow among regions, and ecological efficiency and fiscal expenditure have spatial spillover effects; building a spatial measurement model has become the main method to explore green innovation and other influencing factors [27]. There is financial competition, imitation, and interaction among local governments in China. Therefore, spatial factors should be taken into account when analyzing China's financial expenditure behavior [28].

Most research on ecological efficiency focuses on the distortions of factor prices [29–31], environmental regulation [32], industrial structure [33,34], market segmentation [35], etc. There is less attention paid to how setting economic growth targets impacts ecological efficiency. With clear, simple, and comprehensive characteristics, the economic growth target has become one of the most favored target governance methods of governments at all levels [36]. Economic growth targets in developing countries such as China have an impact on economic growth and environmental pollution.

Throughout the existing literature, studies have primarily focused on the relationship between economic growth target setting and economic growth and environmental pollution. However, there is still room for further exploration, such as the lack of research on ecological efficiency by economic growth target. Second, there is a lack of research on the impact mechanism of economic growth targets on ecological efficiency. This article may contribute in the following ways: from the perspective of research, through the research on local government economic growth targets and their effects on ecological efficiency, the relevant research on ecological efficiency may be enriched; at the level of research data, most of the energy conservation and environmental protection expenditures used in the existing research are provincial data. At present, it is difficult to obtain the energy conservation and environmental protection expenditure data of prefecture-level cities in the public database. By applying to the municipal finance bureaus and statistics bureaus, we have obtained the fiscal energy conservation and environmental protection expenditures of prefecture-level cities, which has refined the research scale; at the level of mechanism exploration, there is a lack of research on the mechanism of the role of economic growth target setting on ecological efficiency in the existing research. Taking the expenditure structure as the intermediate mechanism of the local government's economic growth target affecting ecological efficiency, the research on the mechanism of urban ecological efficiency has widened, which is also instructive for the improvement of ecological efficiency.

## 3. Research Hypothesis

Target governance contains a wealth of Chinese government behavior codes. Economic growth targets represent the will and behavioral basis of local governments to regulate the economy. Over the years, GDP growth has become the best indicator for officials to highlight their achievements [37]. The policy orientation of local governments is affected by the GDP growth targets, and the impact of different levels of GDP growth targets on ecological efficiency is different. Under the setting of the target of moderate economic growth, the local government expects that it will reasonably arrange fiscal expenditure

without stimulating economic growth through large-scale investment [38]. Policy attention will be placed on the provision of productive services, and appropriate economic growth target setting can enable local governments to pay more attention to the quality of economic growth rather than the speed of economic growth, thus helping to improve ecological efficiency. Excessive economic growth targets will inhibit the improvement of ecological efficiency. First, excessive economic growth target setting will lead to repeated construction and performance engineering problems caused by the promotion game in the GDP promotion of local governments. When setting economic targets, local officials will magnify economic growth targets for performance competition, resulting in a series of environmental and economic problems [39]. Local governments expect to increase investment in order to achieve the economic growth target and tend to invest in infrastructure construction, causing problems of repeated economic construction and overcapacity, squeezing out innovation expenditure, which is not conducive to high-quality development of the local economy. Secondly, under the pressure of excessively high economic growth targets, local governments will disrupt enterprise innovation activities. Enterprises follow the investment signals released by local governments, pay attention to short-term economic growth, reduce the level of R&D investment, and inhibit the innovation activities of micro enterprises [40,41], which are not conducive to the improvement of ecological efficiency. Finally, the setting of excessively high economic growth targets makes the local government attract highly polluting enterprises to the local area by reducing the tax burden of enterprises, resulting in an increase in the level of local environmental pollution, which ultimately leads to the setting of excessively high economic growth targets and inhibits the improvement of ecological efficiency. Based on this, the research hypothesis is put forward:

**Hypothesis 1.** *Moderate economic growth targets can improve ecological efficiency, while excessive economic growth targets will inhibit the improvement of ecological efficiency, that is, the relationship between economic growth target setting and ecological efficiency is in an inverted U shape.*

In China, local governments have a variety of development targets, which can be roughly divided into two categories: economic growth and improvement of people's livelihood. The promotion and appointment system of officials with Chinese characteristics plays a decisive role in the selection of local government expenditure. Which type of expenditure local governments prefer depends on the policy guidance of the superior government [19].

Fiscal science and technology expenditure and energy conservation and environmental protection expenditure are characterized by a wide range of benefits, strong externalities, and slow economic returns. Economic expenditure is conducive to the local government to strive for the inflow of factors and rapidly promote economic development. However, the fiscal expenditure on science and technology and the expenditure on energy conservation and environmental protection are typical non-economic public goods investments, which cannot bring about economic growth in a short period of time, which is contrary to the purpose of officials to achieve short-term economic growth and political promotion. At the micro level, fiscal expenditure on science and technology can help strengthen innovation and enhance the enthusiasm of the market for innovation, and expenditure on energy conservation and environmental protection can reduce the fiscal constraints for enterprises to deal with environmental pollution. At the macro level, fiscal science and technology expenditure can promote the coordination of industrial structure and employment structure and improve the efficiency of factor allocation [42].

When the economic growth target setting is combined with the limited tenure of officials, the local government will sort the expenditure, leading to the existence of a bias or correction mechanism for the fiscal expenditure structure in the economic growth target setting. If the target level of economic growth is set too high, local officials will be promoted in order to complete the tasks of their superiors, which will help them pay more economically [43]. Public expenditure affects green development by affecting the quality of the

ecological environment and economic growth. Local governments have limited resources. When resources are used for economic construction, it is conducive to economic growth, but local governments may affect the improvement of ecological efficiency by reducing fiscal and scientific expenditure and energy conservation and environmental protection expenditure. Because local governments have the incentive to develop the economy and obtain political promotion, they will ignore the provision of public goods, which will lead to local governments reducing such input, resulting in the competitive effect of public goods and services, such as repeated infrastructure construction and resource mismatch, which is not conducive to the improvement of ecological efficiency. When the level of economic target setting is relatively reasonable, local governments will give priority to public service expenditure, increase non-productive expenditure, give more consideration to the needs of the people in the area under their jurisdiction, attach importance to the expenditure to increase regional welfare, and increase the proportion of fiscal science and technology expenditure and energy conservation and environmental protection expenditure. Based on this, research hypothesis 2 is put forward:

**Hypothesis 2.** *The economic growth target affects the ecological efficiency by affecting the proportion of energy conservation and environmental protection expenditure and the proportion of fiscal science and technology expenditure.*

## 4. Methods and Data

### 4.1. EBM-DEA Model

The methods for measuring efficiency mainly include parametric method and nonparametric method. Since the nonparametric method does not need to set specific function forms, it is more suitable for efficiency measurement of multiple inputs and outputs, and it is now more widely used. The nonparametric method includes the radial measurement method and the non-radial measurement method. The radial measurement method assumes that the input shrinks in the same proportion, but in reality, different input variables have different elasticity to output variables, and the reduction of input variables is not in the same proportion. Therefore, non-radial measurement methods are more widely used. The non-radial measurement method is mainly SBM-DEA method, but this method loses the proportion information of the projection value of the efficiency frontier. The EBM-DEA model proposed by Tone can effectively solve the shortcomings of radial measurement and non-radial measurement [39]. In this paper, MaxDEA Ultra7.6.1 software is used to analyze the input and output indicators of ecological efficiency. Comprehensive technical efficiency (Crste), pure technical efficiency (Vrste), and scale technical efficiency (Se) are obtained. The connotation of comprehensive technical efficiency indicators is richer. This paper uses comprehensive technical efficiency to measure ecological efficiency.

Suppose that there are $K(k = 1, 2, \ldots K)$ decision-making units in the production process, and each decision-making unit has $N(n = 1, 2, \ldots N)$ inputs and $M(m = 1, 2, \ldots M)$ desirable outputs, the input matrix and desirable output matrix are: $X = \{x_{nk}\} \in R^{N \times K}$, $Y = \{y_{mk}\} \in R^{M \times K}$, and $X > 0, Y > 0$. In the production process, undesirable outputs will inevitably occur. Suppose there are $J(j = 1, 2, \ldots J)$ undesirable outputs in the production process, expressed as $B = b_{jk} \in R^{J \times K}$. With reference to the environmental DEA technology proposed by Faere et al. [36], build an EBM model considering undesirable output:

$$\delta^* = \min \left( \theta - \varepsilon \sum_{n=1}^{N} \frac{\omega_n^- s_n^-}{x_{n0}} \right) \tag{1}$$

$$s.t. \sum_{k=1}^{K} \lambda_k x_{nk} + s_n^- = \theta x_{n0} (n = 1, 2, \ldots N) \tag{2}$$

$$\sum_{k=1}^{K} \lambda_k y_{mk} \geq y_{m0} (m = 1, 2, \ldots M) \tag{3}$$

$$\sum_{k=1}^{K} \lambda_k b_{jk} = b_{j0} (j = 1, 2, \ldots J) \tag{4}$$

$$\lambda_k \geq 0, s_n^- \geq 0 \tag{5}$$

$$\sum_{n=1}^{N} \omega_n^- = 1 \left( \omega_n^- \geq 0, n = 1, 2, \ldots N \right) \tag{6}$$

$\delta^*$ refers to the ecological efficiency considering the undesirable output. The closer the value is to 1, the higher the ecological efficiency is. $\omega_n^-$ represents the weight of the input $n$, $\theta$ represents the efficiency value calculated by the radial model, and $s_n^-$ represents the non-radial relaxation variable of input factors. $\varepsilon$ represents the key parameter combining radial and non-radial relaxation, and when $\varepsilon = 0$, EBM is converted to CCR model. When $\theta = \varepsilon = 1$, the EBM model was converted to a non-radial SBM model.

*4.2. Moran Index*

In reality, cities' economic activities are not closed, and there are flows of material resources and production factors between different cities, so they have spatial relevance. Environmental pollution has a strong spatial spillover, so this paper believes that different cities' ecological efficiency has spatial relevance. Spatial autocorrelation should be tested before spatial econometric analysis, usually using Moran index. The Moran index is usually between $[-1, 1]$. If the Moran index is greater than 0, it indicates that there is a positive spatial correlation. If the Moran index is less than 0, it indicates that there is a negative spatial correlation. The above two cases indicate that the spatial measurement model can be used. If Moran index is 0, it indicates that spatial measurement model is not suitable. In order to judge the spatial correlation of ecological efficiency among regions, this paper uses geographic adjacency matrix, economic distance matrix, and inverse distance matrix to test Moran's I statistics.

$$I_i = \frac{y_i - \overline{y}}{\frac{1}{n} \sum (y_i - \overline{y})^2} \sum_{j \neq i}^{n} \omega_{ij}(y_i - \overline{y}) \tag{7}$$

Spatial correlation analysis. The connotation of ecological efficiency includes economic growth and environmental pollution, so there may be spillover effects of ecological efficiency among regions. Therefore, this paper first observes whether ecological efficiency has spatial correlation. First, we use the global spatial correlation to test whether the ecological efficiency has the spatial spillover effect. It is particularly important to select an appropriate spatial weight matrix for spatial metrological analysis, because the spatial weight matrix represents the dependency between spatial units.

$$I = \frac{n \sum_{i=1}^{n} \sum_{j=1}^{n} \omega_{ij}(y_i - \overline{y}) \left( y_j - \overline{y} \right)}{\left( \sum_{i=1}^{n} \sum_{j=1}^{n} \omega_{ij} \right) \sum_{i=1}^{n} (y_i - \overline{y})^2} \tag{8}$$

The improvement of ecological efficiency level in adjacent areas may also lead to the improvement of ecological efficiency in this area. The first is the geographical adjacency matrix. If there is a common boundary between two regions, the value is 1; otherwise, the value is 0. In order to investigate the geographical and economic correlation between regions, this paper uses geographic adjacency matrix, inverse distance matrix, and economic distance weight as the weighting matrix in the benchmark regression. The formulas use geographic adjacency matrix and economic distance matrix. Therefore, this paper will construct three spatial weight matrices. The first is the geographical adjacency matrix. If two cities are adjacent, 1 is taken; otherwise, 0 is taken.

$$\omega_{ij} = \begin{cases} 1 \; i \; and \; j \; are \; adjacent \\ 0 \; i \; and \; j \; are \; not \; adjacent \end{cases} \tag{9}$$

The second kind of spatial matrix is spatial inverse distance matrix. The correlation of ecological efficiency between regions is closely related to the distance between regions. The closer the distance is, the greater the possibility of being affected by technology spillovers and environmental pollution in nearby regions. In this paper, the reciprocal of geographical distance between cities is taken as the weight for standardization, and the longitude and

latitude coordinates of each city center are taken from China National Basic Geographic Information System.

$$\omega_{ij} = \begin{cases} \frac{1}{d_{ij}} & i \neq j \\ 0 & i = j \end{cases} \tag{10}$$

The third is the economic distance matrix. The economic distance matrix depicts the size of the economic gap between two regions, that is, the distance within the economic space, where $g_i$ and $g_j$ are the average GDP of regional $i$ and regional $j$ from 2007 to 2019, respectively. In the matrix, if the gap between the two regions' real GDP averages is smaller, the economic distance is closer.

$$\omega_{ij} = \begin{cases} \frac{1}{|g_i - g_j|} & i \neq j \\ 0 & i = j \end{cases} \tag{11}$$

*4.3. Spatial Durbin Model*

When discussing the effect of economic growth on ecological efficiency, spatial econometric model is introduced. Therefore, ecological efficiency is taken as the explanatory variable, and economic growth target and its square are taken as the explanatory variable to discuss its impact on ecological efficiency. Considering that the ecological efficiency and government behavior studied in this paper are highly spatial-dependent, and the improvement of ecological efficiency is cumulative, the ecological efficiency of a region in the next year is affected by the ecological efficiency of the previous year. China's top leaders decide the promotion of local leaders through economic growth. Driven by the strategic interaction of local leaders, cities' investment has a spatial effect [44]. Therefore, the first order lag term is introduced into the spatial Durbin model to build the spatial Durbin model:

$$crste_{it} = \alpha + \rho W \times crste_{it} + \beta_1 gdpgoal_{i,t-1} + \beta_2 Wgdpgoal_{i,t-1} + \beta_3 gdpgoal_{i,t-1}^2 + \beta_4 Wgdpgoal_{i,t-1}^2 + \sum \eta X_{it} + \sum \lambda W \times X_{it} + \mu_i + v_t + \varepsilon_{it} \tag{12}$$

*crste* represents ecological efficiency, *gdpgoal* represents economic growth target, $X$ represents a series of control variables, subscripts $i$ and $t$ represent region and year, respectively, $W$ represents spatial weight matrix, $\mu_i$ and $v_t$ represent unobservable regional and temporal fixed effects, and $\varepsilon_{it}$ is a random error term.

*4.4. Variables Measurements*

4.4.1. Economic Growth Target

The core explanatory variable is the economic growth target. The economic growth target published in the collected municipal government work report is taken as the core explanatory variable. The excessive economic growth target is expressed by the square of the economic growth target. About 300 prefecture-level cities in China will set economic growth targets every year and publish them to the public in local government work reports. The GDP growth target data in this paper are collected manually from the annual Government Work Report, the statistical yearbook of prefecture-level cities, and the "special" column of the yearbook published on the portal website of the municipal people's governments at all levels. The reports that cannot be obtained through public information are obtained through the government information disclosure platform. It should be noted that not every year's government work report will report specific target values. For the economic target values expressed in terms of growth intervals, interval mean values will be used instead. For the figures published in terms of total GDP, the desirable GDP growth target is calculated by the formula of (total desirable GDP minus total GDP of last year)/total desirable GDP. Refer to Chai and other measurement methods for setting excessively high economic growth targets, expressed in target square [45].

### 4.4.2. Ecological Efficiency

The dependent variable of this paper is ecological efficiency. Referring to the research of Lin et al. [46], the input indicators mainly include capital input, labor input, and resource input [47,48]. At the level of capital investment, the academia widely uses the fixed asset investment of the whole society to measure the amount of investment in the current year. This paper also follows this approach. In order to ensure the comparability of investment data in different periods, the investment data over the years will be reduced according to the price in the base period, with 2007 as the base period. In terms of labor input, this paper uses the number of employees in the whole society at the end of the year to measure labor input. In terms of resource input, as electricity consumption has become the main form of energy consumption in China, it is a better choice to take electricity consumption as energy input, so the paper chooses the total electricity consumption and water consumption of the whole society to express resource input; at the level of desirable output, 2007 is taken as the base period, and the real GDP of each city without price factor is used to measure it; at the level of undesirable output, in view of China's "double carbon targets" of achieving carbon peak by 2030 and carbon neutralization by 2060, carbon dioxide is included in the evaluation system as an undesirable output [6]. Therefore, in this paper, five indicators including industrial wastewater discharge, industrial sulfur dioxide discharge, industrial smoke (dust) discharge [7], and non-comprehensive utilization rate of solid waste and carbon dioxide discharge are selected as the undesirable outputs, and the entropy method is used to calculate the comprehensive index of environmental pollution as the undesirable output.

### 4.4.3. Fiscal Expenditure

The superior government takes energy conservation and environmental protection expenditure and scientific and technological expenditure as direct assessment indicators, among which energy conservation and environmental protection expenditure is the most direct factor affecting green economic development. At the level of expenditure data, it is difficult to obtain the expenditure data of energy conservation and environmental protection of prefecture-level cities in the public information. The research on this expenditure is carried out at the provincial level at multiple levels, and there is a lack of large sample and long-term data investigation at the prefecture level. In this paper, the fiscal bureau and statistics bureau of each city apply for information disclosure to obtain the fiscal expenditure data on energy conservation and environmental protection and expand the empirical results to the city level.

### 4.4.4. Control Variables

There are many factors that affect the level of regional ecological efficiency, which can be divided into economic factors, structural factors, institutional factors, capital factors, and demographic factors based on the research of Han et al. [49]. According to the existing economic theory, economic development level (Econ) is measured by per capita GDP [29,35]. The industrial structure is a direct factor affecting the ecological environment, which is expressed by the proportion of secondary industry (Stru2), and it is measured by the ratio of secondary industry of each city and the proportion of the annual GDP. The second is the institutional factor. The environmental decentralization in the institutional factor will affect the ecological environment protection and the improvement of ecological efficiency. The environmental decentralization (ED) gives local governments more power to impose environmental administrative penalties, which can mobilize the enthusiasm of local environmental protection departments and improve the environmental quality [50]. The level of environmental decentralization refers to the measurement method of Bai et al. [51]. According to Feng et al., cultural and educational investment can significantly promote ecological civilization, enhance regional innovation ability, and promote regional economic development [52]. Therefore, this paper believes that cultural and educational levels can improve ecological efficiency, measured by the number of regional college students. The

third is the capital factor. Financial development can promote green development and improve the level of ecological efficiency by improving risks and optimizing resource allocation [53]. The financial development level is measured by the ratio of loan balance to deposit balance. Foreign direct investment (FDI) can affect ecological efficiency through technology spillovers and industrial linkages. It is an important engine to promote China's economic growth. Its impact on China's ecological environment has two contradictory effects—the pollution haven effect and the pollution halo effect. At a low level of economic development, foreign direct investment has a negative impact on the ecological environment. As the level of economic development exceeds a certain threshold, foreign direct investment will help improve the ecological environment [54], measured by the proportion of foreign direct investment in GDP. Fourth is the population factor. Cities with high population density can effectively improve the efficiency of sewage disposal [55], and the level of skill complementarity among the working population is high, which can improve the level of resource allocation and improve the ecological efficiency. It is expressed by cities' population density (Popden) and measured by the ratio of cities' area to the total cities' population.

### 4.5. Data

The data used in this paper are mainly from *China City Statistical Yearbook*, *China Fiscal Yearbook*, work reports of municipal governments, statistical bulletins of national economic and social development of cities, EPS database, and wind database. In the selection of sample cities, there were some changes in the administrative level of some cities within the time frame of the study, for example, Anhui Province abolishing Chaohu City in 2011, Guizhou Province withdrawing Bijie and Tongren from districts and cities in 2011, and Shandong Province abolishing Laiwu City in 2019 and classifying it as Jinan City. Due to the changes of these city levels, the above cities and cities with more missing original data are excluded in this paper. The sample data include 284 cities, and the sample period is set as 2007–2019.

Figure 1 is the research framework of this paper. First, the EBM-DEA model is used to measure the ecological efficiency of 284 cities in China from 2007 to 2019. Then, the ArcGIS technology is used to depict the spatial distribution of cities' ecological efficiency. The global spatial autocorrelation and local spatial autocorrelation are used to test the spatial correlation of ecological efficiency. If the test results support the spatial correlation of ecological efficiency, the spatial econometric model is used to explore the impact of economic growth targets on ecological efficiency. Otherwise, OLS regression is used. Finally, the fiscal expenditure on science and technology and environmental protection expenditure are used as intermediary variables to test the hypothesis that economic growth targets affect ecological efficiency through fiscal behavior.

To reflect the social and economic development of different regions, the State Council of China has divided China's economic regions into four: the east, the middle, the west, and the northeast (Figure 2). Among them, the eastern region includes Beijing, Tianjin, Hebei, Shanghai, Shandong, Jiangsu, Zhejiang, Fujian, Guangdong, and Hainan; the central region includes Shanxi, Anhui, Henan, Jiangxi, Hunan, and Hubei; the western region includes Inner Mongolia, Guangxi, Chongqing, Sichuan, Guizhou, Yunnan, Tibet, Shaanxi, Gansu, Qinghai, and Ningxia; Northeast China includes Liaoning, Jilin, and Heilongjiang.

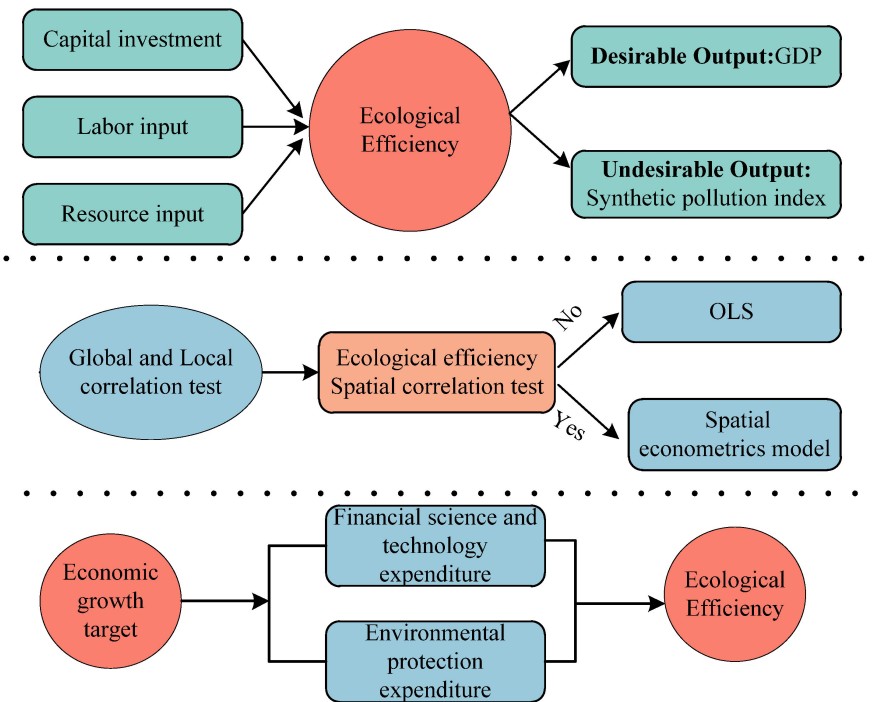

**Figure 1.** Analysis framework.

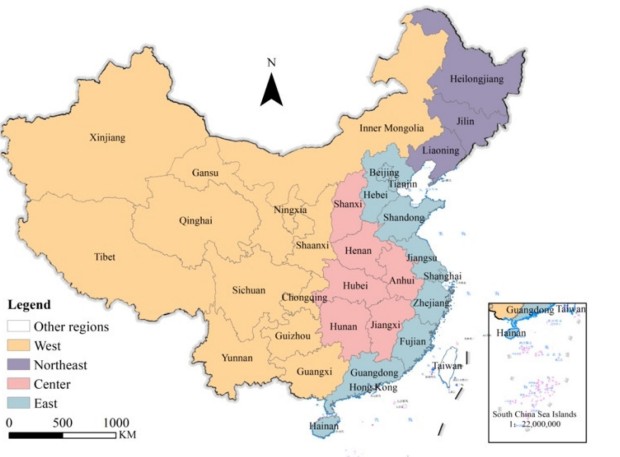

**Figure 2.** Division of four regions in China. Note: the map is drawn according to the standard map service website of the Ministry of Natural Resources (map review no. GS (2020) 4630).

## 5. Empirical Results

### 5.1. Spatial Correlation

This paper first describes the regional distribution characteristics of ecological efficiency using ArcGIS10.8. Then, we use the global spatial autocorrelation and local spatial autocorrelation methods to test the spatial correlation of ecological efficiency.

#### 5.1.1. Spatial and Temporal Distribution of Ecological Efficiency

ArcGIS was used to map the spatial and temporal distribution of ecological efficiency in 2007, 2010, 2015, and 2019 (Figure 3). In order to better compare the changes of ecological efficiency, this paper classifies the ecological efficiency. Because the natural breakpoint method of ArcGIS is used, which makes different time points incomparable, this paper classifies the ecological efficiency by combining the natural breakpoint method and the equal division method. The ecological efficiency is divided into five types, that is, strongly

ineffective (0, 0.4), ineffective (0.4, 0.6), weakly ineffective (0.6, 0.8), weakly effective (0.8, 1.0), and effective (1.0, 1.2). It can be seen from Figure 3 that the level of ecological efficiency of cities in China is rising and showing differentiation. In terms of time dimension, the level of ecological efficiency of cities is basically rising. Among them, the number of cities with strongly ineffective ecological efficiency decreased significantly, from 217 in 2007 to 107 in 2019. The number of cities with ineffective, weakly ineffective, and weakly effective ecological efficiency increased year by year. The number of cities with ineffective ecological efficiency decreased from 217 in 2007 to 107 in 2019. The number of cities with weakly ineffective ecological efficiency increased from 5 in 2007 to 36 in 2019, from 1 in 2007 to 6 in 2019, and from 0 in 2007 to 16 in 2019. In terms of spatial dimension, the ecological efficiency levels of Chinese cities are distributed differently, showing the characteristics of high in the east and low in the west as a whole. The cities with ineffective ecological efficiency are mainly distributed in the central and western regions. The weakly ineffective ecological efficiency has changed from point distribution to block distribution and is mainly concentrated in urban agglomerations in the middle reaches of the Yangtze River, the Yangtze River Delta, and the Central Plains. The cities with weakly effective ecological efficiency are mainly distributed in dots, mainly in eastern China. The effective ecological efficiency is mainly distributed in coastal cities in eastern China. It can be seen from Figure 2 that the gap between cities with high ecological efficiency and cities with low ecological efficiency is gradually widening. Therefore, it is very important to find out the factors that affect the ecological efficiency of each city, continuously improve the ecological efficiency of each city, and narrow the gap between regions.

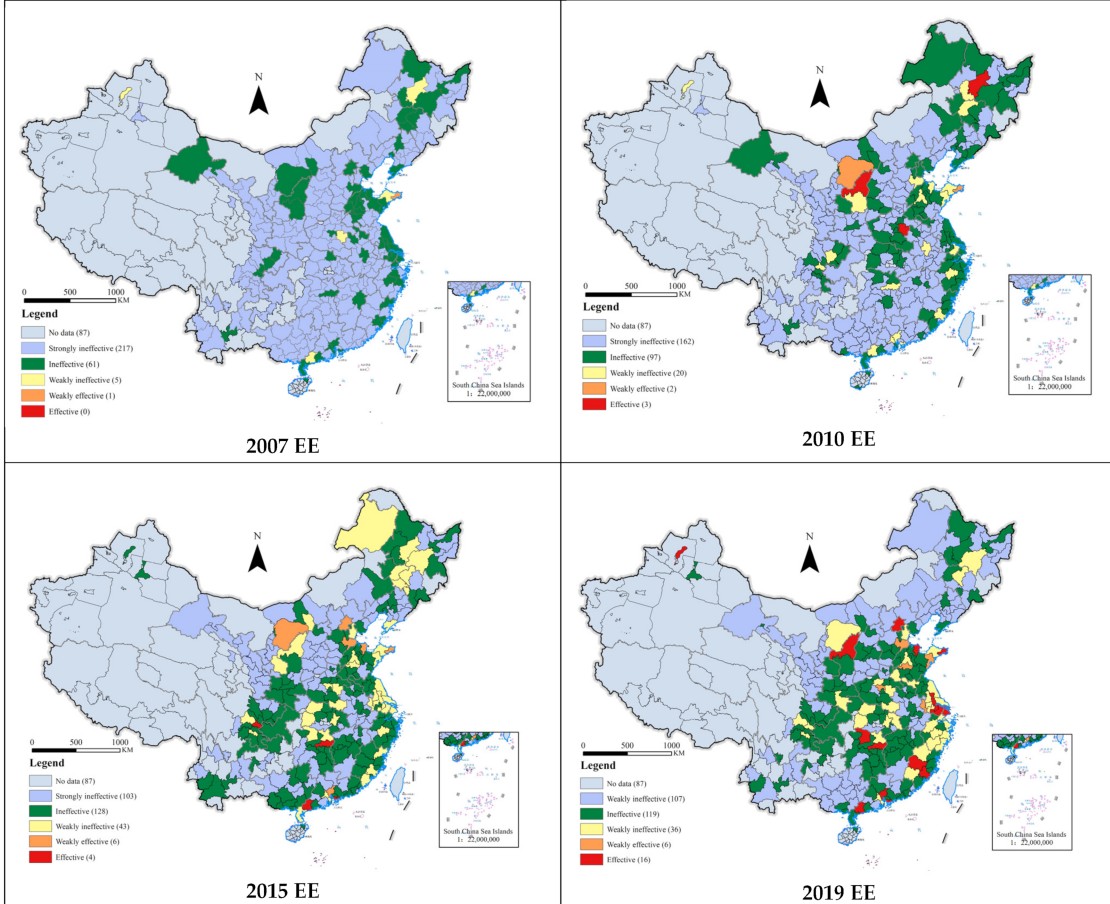

**Figure 3.** Urban ecological efficiency of Chinese cities in 2007, 2010, 2015, and 2019. Note: this map is based on the standard map with the drawing review number of GS (2020) 4619 on the standard map service website of the Ministry of Natural Resources. There is no modification to the base map.

5.1.2. Global Spatial Autocorrelation

Under the geographic adjacency matrix, inverse distance matrix, and economic distance matrix (Table 1), the Moran index is significantly positive, indicating that there is a positive spatial correlation in ecological efficiency, namely high–high concentration and low–low concentration. Specifically, under the geographical adjacency matrix, the Moran index of ecological efficiency is the largest, and the Moran index shows a fluctuating upward trend, indicating that the phenomenon of regional ecological efficiency agglomeration is increasingly significant over time.

**Table 1.** Moran index.

| | Geographic Adjacency Matrix | | Inverse Distance Matrix | | Economic Distance Matrix | |
|---|---|---|---|---|---|---|
| | **I** | ***p*-Value** | **I** | ***p*-Value** | **I** | ***p*-Value** |
| 2007 | 0.308 *** | 0.000 | 0.146 *** | 0.000 | 0.140 *** | 0.000 |
| 2008 | 0.286 *** | 0.000 | 0.134 *** | 0.000 | 0.153 *** | 0.000 |
| 2009 | 0.263 *** | 0.000 | 0.123 *** | 0.000 | 0.164 *** | 0.000 |
| 2010 | 0.262 *** | 0.000 | 0.116 *** | 0.000 | 0.153 *** | 0.000 |
| 2011 | 0.210 *** | 0.000 | 0.093 *** | 0.000 | 0.135 *** | 0.000 |
| 2012 | 0.207 *** | 0.000 | 0.087 *** | 0.000 | 0.120 *** | 0.000 |
| 2013 | 0.200 *** | 0.000 | 0.096 *** | 0.000 | 0.131 *** | 0.000 |
| 2014 | 0.192 *** | 0.000 | 0.095 *** | 0.000 | 0.170 *** | 0.000 |
| 2015 | 0.223 *** | 0.000 | 0.122 *** | 0.000 | 0.203 *** | 0.000 |
| 2016 | 0.289 *** | 0.000 | 0.160 *** | 0.000 | 0.213 *** | 0.000 |
| 2017 | 0.349 *** | 0.000 | 0.204 *** | 0.000 | 0.275 *** | 0.000 |
| 2018 | 0.396 *** | 0.000 | 0.228 *** | 0.000 | 0.273 *** | 0.000 |
| 2019 | 0.368 *** | 0.000 | 0.218 *** | 0.000 | 0.246 *** | 0.000 |

Note: ***, **, and * are significant at the level of 1%, 5%, and 10%, respectively; the values in parentheses represent standard errors.

Referring to Elhorst's inspection ideas [56], this paper also uses the LM test to test the spatial econometric model (Table 2). It is found that the spatial econometric model basically passes the significance test, indicating that the spatial econometric model is applicable to the analysis of this paper.

**Table 2.** LM Test.

| | Mixed Regression | Fixed Regions | Fixed Time | Fixed Time and Region |
|---|---|---|---|---|
| LM test, no spatial lag | 230.437 *** (0.000) | 427.566 *** (0.000) | 219.962 *** (0.000) | 413.727 *** (0.000) |
| Robust LM test, no spatial lag | 2.188 (0.139) | 22.179 *** (0.000) | 5.391 ** (0.020) | 25.999 *** (0.000) |
| LM test, no spatial error | 287.222 *** (0.000) | 431.013 *** (0.000) | 252.550 *** (0.000) | 402.531 *** (0.000) |
| Robust LM test, no spatial error | 58.973 *** (0.000) | 25.626 *** (0.000) | 37.980 *** (0.000) | 14.803 *** (0.000) |

Note: ***, **, and * are significant at the level of 1%, 5%, and 10%, respectively; the values in parentheses represent standard errors.

5.1.3. Local Spatial Autocorrelation

Geoda software is used to measure the local spatial autocorrelation of ecological efficiency, and ArcGIS is used to make the LISA cluster map of ecological efficiency. Figure 4 is the LISA concentration diagram of ecological efficiency under the geographical adjacency matrix, which can more intuitively reflect the current situation of regional concentration. The Moran index of local spatial autocorrelation is in the range of [−1, 1], and its spatial attributes can be divided into two positive correlation types: high–high (H-H), low–low

(L-L), and two negative correlation types: high–low (H-L), low–high (L-H). High–high agglomeration indicates the spatial agglomeration characteristics of high ecological efficiency regions, while low–low agglomeration indicates the spatial agglomeration characteristics of low ecological efficiency regions. High–low concentration means that the local high ecological efficiency area is surrounded by the adjacent low ecological efficiency area, which is the embodiment of the polarization of ecological efficiency. Low–high concentration means that the local ecological efficiency is low and surrounded by areas with high ecological efficiency, which is the embodiment of the transition to high ecological efficiency. Not significant means that the ecological efficiency does not have the characteristics of spatial agglomeration, and the ecological efficiency is randomly distributed in space. From the perspective of the number change of significant ecological efficiency agglomeration, the number of cities with high ecological efficiency and high agglomeration fluctuated and increased, from 22 in 2007 to 27 in 2019. The number of low–low cluster cities has increased year by year, from 17 in 2007 to 34 in 2019. High–low concentration shows a fluctuating trend, with a small number. The number of low–high cluster cities has been relatively small, showing an upward trend, from 3 in 2007 to 7 in 2019. In terms of spatial distribution, high–high agglomeration is mainly distributed in Harbin Great Wall urban agglomeration, Beijing Tianjin Hebei urban agglomeration, Shandong Peninsula urban agglomeration, Yangtze River Delta urban agglomeration, Pearl River Delta urban agglomeration, and Chengdu Chongqing urban agglomeration. Low–low agglomeration is mainly distributed in central and southern Liaoning urban agglomeration, Beibu Gulf urban agglomeration, Xining Lanzhou urban agglomeration, and central Guizhou urban agglomeration. In the future, we should focus on breaking the low–low ecological efficiency cluster and promoting the increase of the number of high–high urban ecological efficiency clusters.

### 5.1.4. Selection of Measurement Model

First, the variables used in this paper are described and statistically analyzed, as shown in Table 3. Among them, Crste is the comprehensive technical efficiency, which is measured by the EBM-DEA measurement results. Enfis and Tecfis are fiscal expenditure structures, which are measured by the proportion of energy conservation and environmental protection expenditure in local fiscal expenditure and the proportion of science and technology expenditure in total local fiscal expenditure. Stru2 is the proportion of the secondary industry, measured by the ratio of the gross output value of the secondary industry to the total output value. Reasonable target setting for economic growth is measured by ln (Target + 1). Target$^2$ is the target setting of excessive economic growth, measured by ln (Target$^2$ + 1). Popden is the regional population density, which is measured by logarithmic population density. FINLEV is the level of financial development, measured by the proportion of total loans and deposits. FDI is the level of foreign direct investment, measured by the ratio of foreign direct investment to total regional production. HRCAP is the level of human capital, measured by the number of undergraduates and above. ED indicates the level of environmental decentralization, using $\frac{Sys_{it}/Pop_{it}}{Sys_t/Pop_t} \times [1 - (Gdp_{it}/Gdp_t)]$ to measure, where $Env_{it}$ refers to the total number of environmental protection system personnel in city $i$ in year $t$, and $Sys_t$ represents the total number of personnel in the national environmental protection system in year $t$. $Pop_{it}$ refers to the population size of city $i$ in year $t$, and $Pop_t$ represents the total population size of the country in year $t$. $Gdp_{it}$ represents the gross domestic product of city $i$ in year $t$, $Gdp_t$ represents the gross domestic product in year $t$, and $[1 - (Gdp_{it}/Gdp_t)]$ is the scaling factor of economic scale. In the robustness test, green patents and SBM-DEA were used to measure the ecological efficiency.

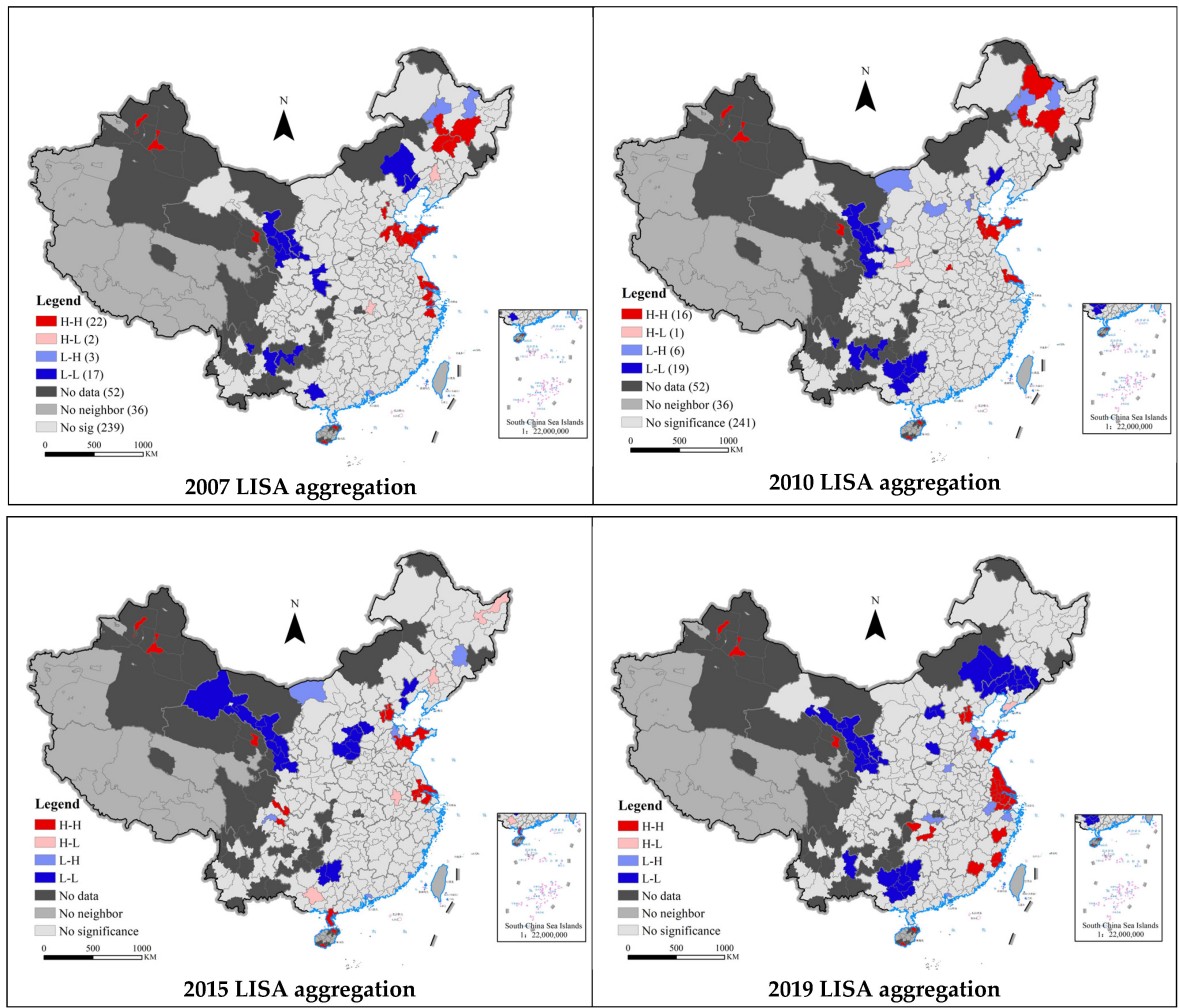

**Figure 4.** LISA cluster of ecological efficiency. Note: this map is based on the standard map with the drawing review number of GS (2020) 4619 on the standard map service website of the Ministry of Natural Resources. There is no modification to the base map.

**Table 3.** Descriptive statistical analysis.

| Varname | Measurement | Obs | Mean | SD | Min | Median | Max |
|---|---|---|---|---|---|---|---|
| Crste | Crste | 3408 | 0.449 | 0.158 | 0.202 | 0.416 | 1.007 |
| Enfis | Environment protection expenditure/local fiscal expenditure | 3408 | 0.030 | 0.017 | 0.000 | 0.027 | 0.151 |
| Tecfis | Technology fiscal expenditure/local fiscal expenditure | 3408 | 0.015 | 0.015 | 0.001 | 0.011 | 0.131 |
| Econ | Ln(per capita GDP) | 3408 | 10.292 | 0.600 | 8.118 | 10.269 | 11.989 |
| Stru2 | Structure2/GDP | 3408 | 0.476 | 0.106 | 0.197 | 0.480 | 0.747 |
| Target | Ln(Target + 1) | 3408 | 2.452 | 0.257 | 0.693 | 2.485 | 3.466 |
| Target$^2$ | Ln(Target$^2$ + 1) | 3408 | 4.728 | 0.559 | 0.693 | 4.804 | 6.869 |
| Popden | Ln Popden | 3408 | 0.381 | 0.267 | 0.025 | 0.287 | 1.505 |
| FINLEV | Loan/Deposit | 3408 | 0.660 | 0.178 | 0.301 | 0.648 | 1.174 |
| FDI | FDI/GDP | 3408 | 0.006 | 0.013 | 0.000 | 0.001 | 0.205 |
| HRCAP | Number of undergraduates and above/100,000 | 3408 | 0.090 | 0.161 | 0.000 | 0.034 | 1.153 |
| ED | $Environ_{it} = \frac{Sys_{it}/Pop_{it}}{Sys_t/Pop_t} \times [1 - (Gdp_{it}/Gdp_t)]$ | 3408 | 0.118 | 0.086 | 0.021 | 0.093 | 0.467 |
| lngpatent | Number of green patents | 3408 | 4.799 | 1.766 | 0.000 | 4.710 | 10.507 |
| sbmcrste | Ecological efficiency measured by SBM-DEA method | 3408 | 0.414 | 0.157 | 0.150 | 0.382 | 1.081 |

After determining the spatial measurement model, it is necessary to determine whether to use the spatial lag model, the spatial error model, or the spatial Durbin model. First, LM and LR tests are used to determine which spatial econometric model to use, and then Hausman tests are used to determine whether to use a fixed effect model or random effect model (Table 4).

**Table 4.** LM, LR, and Hausman inspection.

| Test | Geographic Adjacency Matrix | | Inverse Distance Matrix | | Economic Distance Matrix | |
|---|---|---|---|---|---|---|
| | Statistic | *p*-Value | Statistic | *p*-Value | Statistic | *p*-Value |
| LM spatial error | 641.562 | 0.000 | 5.978 | 0.000 | 1.065 | 0.002 |
| LM spatial autocorrelation | 200.676 | 0.000 | 31.800 | 0.000 | 1.034 | 0.000 |
| LR test SDM SAR | 42.020 | 0.000 | 255.340 | 0.000 | 15.695 | 0.000 |
| LR test SDM SEM | 93.410 | 0.000 | 5920.28 | 0.000 | 154.374 | 0.000 |
| Hausman | 136.810 | 0.000 | 235.657 | 0.000 | 175.768 | 0.000 |

First, the LM test is used to find that under the three weight matrices, the *p*-value of the spatial error model is significant at 1% level, indicating that the SEM model can be used. The *p*-value of the spatial autocorrelation model is also significant at the level of 1%, indicating that the SAR model can be selected, so both the SEM model and SAR model are appropriate, so we chose the SDM model, combining the two. Using the LR test, under the three weight matrices, the *p* value of the spatial error model is significant at the 1% confidence level, indicating that the spatial Durbin model cannot be degenerated into a spatial error model. The *p* value of the spatial lag model is also significant at the 1% confidence level, indicating that the spatial Durbin model cannot degenerate into the spatial autocorrelation model, so the spatial Durbin model is selected. Finally, according to the results of Hausman test, it was found that the spatial Durbin model with fixed time and fixed individual is more appropriate.

*5.2. Benchmark Model*

Considering that the impact of economic growth target setting on ecological efficiency is lagging, the article will lag the variable economic growth target by one period to explore its role in ecological efficiency. The time range of the explained variable in the benchmark regression is 2008–2019, and the time range of the explanatory variables is 2007–2018. Due to the strong path dependence of ecological efficiency, the ecological efficiency of local governments this year is affected by the ecological efficiency of the previous year. In order to control the dynamic spatial panel regression of this year, the first-order lag term is introduced on the basis of the spatial Durbin model, and Stata15.1 software is used to explore the relationship between economic growth targets and ecological efficiency by using the dual fixed spatial Durbin model and the dynamic dual fixed spatial Durbin model. The results are reported in Table 5.

**Table 5.** Benchmark regression.

| | Geographic Adjacency Matrix | | Inverse Distance Matrix | | Economic Distance Matrix | |
|---|---|---|---|---|---|---|
| | Double Fixed Effect Spatial Durbin Model | Dynamic Double Fixed Effect Spatial Durbin Model | Double Fixed Effect Spatial Durbin Model | Dynamic Double Fixed Effect Spatial Durbin Model | Double Fixed Effect Spatial Durbin Model | Dynamic Double Fixed Effect Spatial Durbin Model |
| | Main | | Main | | Main | |
| L.Crste | | 0.734 *** | | 0.763 *** | | 0.750 *** |
| | | (0.014) | | (0.014) | | (0.014) |
| L.Target | 0.361 * | 0.892 *** | 0.212 | 0.988 *** | 0.451 * | 0.859 *** |
| | (0.263) | (0.219) | (0.260) | (0.218) | (0.258) | (0.211) |
| L.Target $^2$ | −0.185 * | −0.400 *** | −0.115 | −0.449 *** | −0.229 * | −0.387 *** |
| | (0.120) | (0.100) | (0.119) | (0.100) | (0.118) | (0.097) |
| Econ | 0.195 *** | 0.088 *** | 0.220 *** | 0.091 *** | 0.229 *** | 0.095 *** |
| | (0.013) | (0.012) | (0.013) | (0.012) | (0.011) | (0.010) |
| Stru2 | 0.003 | 0.015 | −0.013 | 0.008 | 0.001 | 0.032 |
| | (0.034) | (0.030) | (0.034) | (0.030) | (0.033) | (0.029) |
| Popden | 0.010 | 0.004 | 0.009 | −0.006 | 0.021 ** | 0.009 |
| | (0.008) | (0.007) | (0.008) | (0.007) | (0.008) | (0.007) |
| FDI | 1.423 *** | 0.403 *** | 1.455 *** | 0.423 *** | 1.364 *** | 0.354** |
| | (0.179) | (0.149) | (0.178) | (0.149) | (0.184) | (0.150) |
| FINLEV | −0.075 *** | −0.063 *** | −0.059 *** | −0.044 *** | −0.067 *** | −0.067 *** |
| | (0.016) | (0.014) | (0.016) | (0.014) | (0.016) | (0.013) |
| HRCAP | 0.286 *** | 0.115 *** | 0.280 *** | 0.105 *** | 0.062 | 0.039 |
| | (0.043) | (0.038) | (0.043) | (0.038) | (0.047) | (0.041) |
| ED | 0.011 | 0.032 | −0.005 | 0.034 | 0.033 | 0.050 ** |
| | (0.030) | (0.025) | (0.030) | (0.025) | (0.031) | (0.025) |
| L.Target·W | −0.037 *** | −0.131* | −1.319 | −0.308 | 0.115* | 0.664 |
| | (0.007) | (0.108) | (0.936) | (0.774) | (0.060) | (0.558) |
| L.Target$^2$ ·W | 0.015 | 0.066 | 0.595 | 0.080 | −0.039 | −0.295 * |
| | (0.224) | (0.187) | (0.428) | (0.354) | (0.312) | (0.155) |
| Econ·W | −0.059 *** | −0.047 *** | −0.213 *** | −0.393 *** | −0.084 *** | −0.055 * |
| | (0.019) | (0.017) | (0.037) | (0.032) | (0.032) | (0.029) |
| Stru2·W | 0.055 | 0.107 ** | 0.173 | 0.070 | 0.136 | 0.122 |
| | (0.056) | (0.049) | (0.117) | (0.101) | (0.094) | (0.080) |
| Popden·W | 0.070 *** | 0.022 | 0.115 *** | −0.124 *** | 0.049 ** | 0.017 |
| | (0.016) | (0.014) | (0.038) | (0.033) | (0.024) | (0.021) |
| FDI·W | 0.325* | 0.175 *** | 0.602 | 0.917 | 2.202 *** | 1.454 ** |
| | (0.163) | (0.010) | (0.838) | (0.704) | (0.737) | (0.594) |
| FINLEV·W | 0.029 | −0.020 | −0.088 | 0.071 | 0.099 ** | 0.009 |
| | (0.026) | (0.022) | (0.060) | (0.052) | (0.042) | (0.035) |
| HRCAP·W | 0.014 | 0.031 | 0.063 | 0.167 | 1.596 *** | 0.457 *** |
| | (0.082) | (0.073) | (0.221) | (0.196) | (0.140) | (0.122) |
| ED·W | 0.050 | 0.050 | 0.256 ** | −0.134 | −0.029 | −0.050 |
| | (0.052) | (0.043) | (0.121) | (0.102) | (0.093) | (0.075) |
| Spatial rho | 0.342 *** | 0.163 *** | 0.731 *** | 1.574 *** | 0.137 *** | 0.060 ** |
| | (0.020) | (0.019) | (0.035) | (0.040) | (0.032) | (0.029) |
| N | 3408 | 3124 | 3408 | 3124 | 3408 | 3124 |
| $R^2$ | 0.308 | 0.838 | 0.222 | 0.868 | 0.318 | 0.802 |

Note: ***, **, and * are significant at the level of 1%, 5%, and 10%, respectively; the values in parentheses represent standard errors.

Double fixed effect and dynamic spatial Durbin models are used to test the impact of economic growth targets on ecological efficiency by using the geographic adjacency matrix, inverse distance matrix, and economic distance matrix. It can be seen from the benchmark regression in Table 5 that when the geographical adjacency matrix is used as the weight, the results of the double fixed effect spatial Durbin model and the dynamic double fixed effect spatial Durbin model are roughly the same. In this paper, the geographical adjacency matrix is mainly selected for illustration. From the regression results in Table 5, it can be

seen that the $R^2$ of the dynamic double fixed effect spatial Durbin model is larger, indicating that the latter model has a better fitting effect. It can be seen from the rho value that the ecological efficiency is significantly positive at the 1% confidence level, indicating that there is a positive spatial correlation between the ecological efficiency, that is, high–high clustering and low–low clustering.

This paper mainly chooses the regression results under the geographical adjacency matrix to explain. Specifically, under the geographical adjacency matrix, the value of spatial rho is 0.342 and 0.163 in the main effect regression, which is significant, indicating that the ecological efficiency has a positive spatial correlation. $R^2$ is 0.308 when using the double fixed space Durbin model and 0.838 when using the double fixed dynamic space Durbin model, indicating that the fitting effect is better when using the double fixed dynamic space Durbin model. The impact of reasonable economic growth targets lagging behind by one period on ecological efficiency is significantly positive, with the effect of 0.361 and 0.892, respectively, indicating that the government can effectively improve ecological efficiency by setting reasonable economic growth targets. The ecological efficiency lagging behind by one period has a positive effect on the ecological efficiency of the current period, indicating that the ecological efficiency is path-dependent, and the ecological efficiency of the previous year will affect the ecological efficiency level of the current period. The excessive economic growth target lagging behind by one period setting has a significant negative impact on ecological efficiency, with the effects of −0.185 and −0.400, respectively, indicating that the government's excessive economic growth target setting inhibits the improvement of ecological efficiency. Under the appropriate economic growth target, the local government should reasonably arrange the fiscal expenditure, which is conducive to the efficient allocation of factors. The government's use of financial resources for environmental protection and improvement of innovation capacity is conducive to improving ecological efficiency. Under the excessively high economic growth target, local governments, in order to achieve the economic growth target, tend to invest financial resources in areas that can bring short-term economic growth, such as infrastructure, because environmental protection and science and technology expenditures cannot bring about economic growth in the short term. Although infrastructure can bring short-term economic growth, repeated construction in many places leads to resource mismatch, which inhibits environmental protection expenditure and science and technology expenditure, and it is not conducive to the improvement of ecological efficiency and sustainable development. In the regression of the spatial lag term, it can be seen that the reasonable economic growth target in the adjacent areas has a significant inhibition effect on the local ecological efficiency, and the impact effects are −0.037 and −0.131, respectively. The excessively high economic growth target of neighboring areas can improve the local ecological efficiency, with the impact effects of 0.015 and 0.066, respectively, but it is not significant. According to Li et al., economic growth has a spatial spillover effect [57]. Urban economic growth is affected not only by local growth effects but also by surrounding cities.

The spatial spillover effect of regional ecological efficiency is not only characterized by geographical distance but also affected by the gap of economic development level. If only geographical distance is used to measure the spatial spillover effect of ecological efficiency, there will inevitably be deviation. Therefore, this paper selects the per capita GDP of prefecture-level cities as the matrix element, constructs the economic distance matrix, and carries out spatial regression. The autocorrelation coefficient under the economic distance matrix is positive and significant at the 1% and 5% levels, indicating that the ecological efficiency of a city will be affected by cities with closer economic attributes. The ecological efficiency lagging behind the first stage has a significant impact on the local ecological efficiency, which is 0.750. The reasonable economic target setting lagging behind by one period can effectively improve the local ecological efficiency. Under the two-way fixed effect and dynamic two-way fixed effect models, the size is 0.451 and 0.859, respectively. A reasonable economic target setting can significantly improve the local ecological efficiency, with the effect size of 0.115. Cities with similar economic development levels have an

imitation effect in setting economic growth goals. The reasonable economic target setting lagging behind by one period inhibited the improvement of local ecological efficiency, with the effect of −0.039, but it is not significant. In the dynamic spatial Durbin model, neighboring regions with similar economic development levels lag behind the reasonable economic goal setting of Phase I to promote the improvement of local ecological efficiency, but the effect is not significant. Neighboring regions with similar economic development levels lag behind the excessive economic goal setting of Phase I to significantly inhibit the improvement of local ecological efficiency, and the size is −0.295. Among the control variables, economic development level has a significant positive impact on ecological efficiency, and the effect sizes are 0.195 and 0.088, indicating that the level of local economic development can effectively improve ecological efficiency. The economic development level of geographically adjacent cities significantly inhibited the improvement of local ecological efficiency, with the effect sizes of −0.059 and −0.047. The relatively high level of economic development in neighboring regions has attracted more production factors to flow to the region, leading to the reduction of local ecological efficiency. The impact of the secondary industry on ecological efficiency is not significant, with the magnitude of the impact being 0.003 and 0.015, respectively. Although the secondary industry is mostly an industry that can bring environmental pollution, according to the general law, the technological progress of the secondary industry, especially the manufacturing industry, is significantly faster than that of the service industry, so it is easier to obtain a higher economic growth rate [58], but the secondary industry cannot effectively promote the improvement of local ecological efficiency.

The structure of secondary industry in neighboring areas has a positive effect on local ecological efficiency; under the geographical adjacency matrix, the impact effects of population density on ecological efficiency are 0.010 and 0.004, but this effect is not significant. Foreign investment can significantly improve the ecological efficiency, and the impact effects are 1.423 and 0.403, respectively; in the regression of the spatial lag term, foreign investment in neighboring areas can significantly improve local ecological efficiency. This shows that the impact of foreign investment on ecological efficiency supports the Porter hypothesis. Foreign investment can promote scientific and technological progress, enrich capital, and effectively improve ecological efficiency [59]. Financial development significantly inhibits the improvement of ecological efficiency; the level of human capital can significantly improve the ecological efficiency, and its effect is 0.286. The level of human capital in neighboring areas can promote the local ecological efficiency, but the effect is not significant; Environmental decentralization can improve the local ecological efficiency, but the effect is not significant. Environmental decentralization in adjacent areas cannot significantly improve the local ecological efficiency.

*5.3. Heterogeneity Analysis*

The relatively high level of economic development in neighboring regions has attracted more production factors to flow to the region, leading to the reduction of local ecological efficiency, specifically in the economic basis, innovative resources, environmental pollution level, financial capacity, and other aspects [60]. The formulation of economic growth goals is different, and there must be great differences in the behavior choices of the government. Therefore, the impact of economic growth goals on ecological efficiency may vary from region to region. Therefore, different regions have different incentives for officials when setting economic growth targets, resulting in different impacts on ecological efficiency. Therefore, this paper divides China into four regions, the east, the middle, the west, and the northeast, and explores the relationship between urban economic growth targets and ecological efficiency in different regions (Table 6).

**Table 6.** Heterogeneity regression.

| | Crste | | | | | | | |
| | Geographic Adjacency Matrix | | | | Economic Distance Matrix | | | |
| | East | Central | West | Northeast | East | Central | West | Northeast |
|---|---|---|---|---|---|---|---|---|
| CrsteL1 | 0.858 *** | 0.758 *** | 0.582 *** | 0.512 *** | 0.870 *** | 0.784 *** | 0.589 *** | 0.514 *** |
| | (0.026) | (0.026) | (0.027) | (0.047) | (0.026) | (0.025) | (0.026) | (0.047) |
| L.Target | −4.095 *** | 0.884 | 0.818* | 0.062 | −3.014 *** | 0.996* | 0.671 | 0.242 |
| | (0.921) | (0.544) | (0.483) | (0.395) | (0.883) | (0.518) | (0.481) | (0.380) |
| L.Target $^2$ | 1.866 *** | −0.411 * | −0.368 * | −0.035 | 1.375 *** | −0.468 ** | −0.308 | −0.114 |
| | (0.424) | (0.248) | (0.222) | (0.179) | (0.407) | (0.236) | (0.222) | (0.173) |
| Econ | 0.033 | 0.076 *** | 0.144 *** | 0.150 *** | 0.067 *** | 0.095 *** | 0.134 *** | 0.143 *** |
| | (0.026) | (0.020) | (0.020) | (0.038) | (0.024) | (0.019) | (0.018) | (0.035) |
| Stru2 | −0.123 * | 0.017 | 0.030 | 0.027 | −0.104 | 0.018 | 0.058 | 0.052 |
| | (0.068) | (0.053) | (0.045) | (0.099) | (0.066) | (0.052) | (0.044) | (0.100) |
| Popden | 0.009 | −0.016 | 0.003 | 0.042* | 0.012 | −0.013 | −0.001 | 0.042* |
| | (0.018) | (0.011) | (0.012) | (0.022) | (0.018) | (0.011) | (0.012) | (0.022) |
| FDI | 0.249 | 0.916 ** | 0.080 | 0.496 | 0.191 | 0.699 * | 0.214 | 0.243 |
| | (0.194) | (0.361) | (0.517) | (0.488) | (0.195) | (0.397) | (0.515) | (0.453) |
| FINLEV | −0.079 *** | −0.007 | −0.054 ** | 0.004 | −0.076 *** | −0.017 | −0.064 *** | −0.029 |
| | (0.028) | (0.023) | (0.024) | (0.039) | (0.027) | (0.022) | (0.023) | (0.038) |
| HRCAP | 0.021 | 0.086 | 0.158** | 0.484** | 0.029 | 0.039 | 0.102 | 0.346 |
| | (0.074) | (0.055) | (0.071) | (0.217) | (0.076) | (0.059) | (0.078) | (0.248) |
| ED | 0.063 | −0.096 ** | 0.177 *** | −0.004 | 0.115 ** | −0.124 *** | 0.144 *** | −0.024 |
| | (0.049) | (0.045) | (0.047) | (0.066) | (0.050) | (0.045) | (0.047) | (0.063) |
| L.Target·W | −3.003 ** | 2.173 ** | 0.307 | −0.487 | 0.024 | −0.086 | 0.101 | 0.143 |
| | (1.505) | (0.948) | (0.797) | (0.809) | (2.407) | (1.750) | (0.955) | (1.071) |
| L.Target $^2$·W | 1.364 ** | −1.011 ** | −0.160 | 0.202 | −0.005 | 0.005 | −0.038 | −0.048 |
| | (0.693) | (0.433) | (0.368) | (0.367) | (1.113) | (0.799) | (0.436) | (0.495) |
| Econ·W | 0.012 | 0.068* | 0.015 | −0.058 | 0.024 | 0.056 | 0.007 | −0.125 |
| | (0.043) | (0.038) | (0.025) | (0.064) | (0.062) | (0.067) | (0.050) | (0.085) |
| Stru2·W | 0.077 | 0.006 | 0.015 | 0.307 * | 0.046 | −0.125 | 0.020 | 0.128 |
| | (0.109) | (0.097) | (0.065) | (0.183) | (0.187) | (0.180) | (0.102) | (0.212) |
| Popden·W | 0.052 | −0.012 | 0.034 * | −0.016 | 0.037 | 0.017 | −0.033 | 0.022 |
| | (0.036) | (0.028) | (0.021) | (0.043) | (0.047) | (0.041) | (0.033) | (0.040) |
| FDI·W | −0.283 | −0.177 | 0.703 | −2.160 * | 0.941 | 1.680 | 11.196 ** | 1.891 |
| | (0.383) | (0.653) | (0.952) | (1.121) | (0.870) | (1.516) | (4.983) | (1.262) |
| FINLEV · W | −0.058 | −0.063 | 0.057* | 0.033 | 0.096 | 0.095 | 0.026 | 0.160 * |
| | (0.051) | (0.041) | (0.034) | (0.083) | (0.077) | (0.081) | (0.049) | (0.088) |
| HRCAP·W | −0.008 | 0.112 | 0.001 | 0.186 | 0.388 ** | 0.270 | −0.634 ** | 0.128 |
| | (0.151) | (0.129) | (0.109) | (0.456) | (0.181) | (0.399) | (0.268) | (0.355) |
| ED ·W | 0.173 * | −0.312 *** | −0.017 | 0.107 | 0.276 * | −0.085 | −0.183 | 0.021 |
| | (0.089) | (0.102) | (0.064) | (0.135) | (0.145) | (0.151) | (0.113) | (0.115) |
| Spatial rho | 0.202 *** | 0.055 | 0.013 | 0.071 | 0.019 *** | 0.026 | 0.124 ** | 0.050 |
| | (0.031) | (0.041) | (0.034) | (0.069) | (0.002) | (0.077) | (0.056) | (0.070) |
| N | 946 | 880 | 924 | 374 | 946 | 880 | 924 | 374 |
| $R^2$ | 0.870 | 0.742 | 0.583 | 0.559 | 0.822 | 0.732 | 0.656 | 0.603 |

Note: ***, **, and * are significant at the level of 1%, 5%, and 10%, respectively; the values in parentheses represent standard errors.

In the eastern region, the economic growth target inhibits the improvement of ecological efficiency. Under the geographical adjacency matrix and economic distance matrix, the effect size is −4.095 and −3.014, respectively. An excessively high economic growth target is conducive to the improvement of ecological efficiency. Under the two matrices, the effects are 1.866 and 1.375, respectively. The spatial effect is significant, and the spatial rho values are 0.202 and 0.019, respectively. The eastern region has gradually formed a development mechanism dominated by independent innovation and a new industrial structure in its development. According to the research of Liu et al., the eastern region has the highest innovation index, its innovation index accounts for the largest share of the

national innovation index [61], and it has a large number of scientific and technological talents and strong research funds. In addition, the economic foundation and development speed of the eastern region have always been better than that of the central and western regions, and the impetus to promote economic growth is sufficient. The setting of economic growth targets will not exert pressure on the eastern region, and government behavior will not distort and alienate, so the ecological efficiency has not been significantly affected. In the case of setting the set economic growth target, the local government does not need to distort the input of factors and the total amount and structure of financial expenditure to achieve the economic growth target, so the eastern region sets a high economic growth target to improve the ecological efficiency.

The setting of economic growth targets in the central region, western region, and northeast region can significantly improve ecological efficiency, while the setting of excessive economic growth targets will inhibit ecological efficiency. Compared with the east, the central, western, and northeast regions lag behind in industrialization and economic development. The setting of economic growth target in the central region has a greater impact on ecological efficiency than that in the northeast and western regions. The ecological efficiency in the northeast and western regions has roughly the same impact on economic growth targets. The central region, as the main undertaking place of industries in the eastern region, will evaluate the undertaking industries and undertake the industries that can drive the innovation and rise of the central region when the economic targets are set reasonably. However, if the central region faces too high pressure on economic growth targets, it may undertake enterprises with high pollution and energy consumption, which will lead to ineffective undertaking, ecological threats, decline in natural carrying capacity, and other problems, leading to a sharp decline in ecological efficiency. In recent years, the driving force of economic growth in the western and northeastern regions has gradually shifted from investment and consumption to industrialization-process-driven and loose fiscal-policy-driven forces [62]. Although they have also gradually inherited industries from the eastern region, they lag behind the central region due to economic development foundation and location factors, so the impact of economic growth target setting on the ecological efficiency of the northeast and western regions is lower than that of the central region. To sum up, the ecological efficiency of the central region is most affected by the economic growth targets, followed by the northeast and west.

*5.4. Robust Test*

In order to further verify the robustness of the results, this paper conducts the following robustness tests: the first is to replace the explained variables, according to Effie et al.'s research, and use green patents to measure ecological efficiency [63,64]. The second method uses SBM-DEA to measure ecological efficiency; the third is to exclude municipalities directly under the central government and only retain the sample of prefecture-level cities. From the spatial rho value, the regression result is positive and significant, so the impact of economic growth targets on ecological efficiency has a spatial spillover effect. In addition, $R^2$ is above 0.7, indicating that the fitting effect of several models is good (Table 7).

**Table 7.** Robustness test.

| | Green Patent | | Measured by SBM-DEA | | Exclusion of Municipalities Directly under the Central Government | |
|---|---|---|---|---|---|---|
| | Geographic Adjacency Matrix | Economic Distance Matrix | >Geographic Adjacency Matrix | >Economic Distance Matrix | >Geographic Adjacency Matrix | >Economic Distance Matrix |
| L.lngpatent | 0.496 *** (0.015) | 0.550 *** (0.015) | | | | |
| L.SBM-DEA | | | 0.710 *** (0.015) | 0.720 *** (0.015) | | |
| L.Crste | | | | | 0.724 *** (0.014) | 0.742 *** (0.014) |
| L.Target | 4.002 ** (1.570) | 2.783 * (1.468) | 0.977 *** (0.236) | 0.933 *** (0.226) | 0.856 *** (0.220) | 0.819 *** (0.212) |
| L.Target$^2$ | −14.380 *** (5.316) | −9.559 * (5.096) | −0.437 *** (0.108) | −0.419 *** (0.104) | −0.384 *** (0.101) | −0.368 *** (0.097) |
| Econ | −0.002 (0.081) | 0.088 (0.069) | 0.091 *** (0.013) | 0.094 *** (0.011) | 0.092 *** (0.012) | 0.098 *** (0.010) |
| Stru2 | 0.514 ** (0.210) | 0.665 *** (0.204) | 0.015 (0.032) | 0.037 (0.031) | 0.012 (0.030) | 0.030 (0.029) |
| Popden | −0.043 (0.051) | −0.034 (0.052) | 0.005 (0.008) | 0.011 (0.008) | 0.006 (0.007) | 0.009 (0.007) |
| FDI | −0.548 (1.033) | −0.422 (1.057) | 0.523 *** (0.160) | 0.481 *** (0.161) | 0.317 (0.214) | 0.143 (0.214) |
| FINLEV | 0.042 (0.095) | −0.057 (0.093) | −0.069 *** (0.015) | −0.069 *** (0.014) | −0.060 *** (0.014) | −0.065 *** (0.013) |
| HRCAP | −0.147 (0.267) | −0.026 (0.286) | 0.112 *** (0.041) | 0.029 (0.043) | 0.137 *** (0.040) | 0.069 (0.042) |
| ED | −0.064 (0.175) | −0.060 (0.176) | 0.031 (0.027) | 0.050 * (0.027) | 0.030 (0.025) | 0.045 * (0.025) |
| L.Target·W | −5.539 ** (2.545) | −1.394 (4.269) | 0.174 (0.438) | 0.634 (0.597) | 0.148 (0.410) | 0.656 (0.559) |
| L.Target$^2$ · W | 20.293 ** (8.934) | 4.813 (15.386) | −0.087 (0.201) | −0.279 (0.273) | −0.073 (0.188) | −0.292 (0.256) |
| Econ·W | 0.065 (0.114) | 0.090 (0.199) | −0.048 *** (0.018) | −0.048 (0.031) | −0.035** (0.017) | −0.067 ** (0.029) |
| Stru2·W | 0.442 (0.341) | 0.321 (0.568) | 0.121 ** (0.052) | 0.143 * (0.086) | 0.087 * (0.049) | 0.145 * (0.081) |
| Popden·W | −0.117 (0.098) | −0.383 ** (0.153) | 0.036 ** (0.015) | 0.009 (0.023) | 0.028 * (0.014) | 0.014 (0.021) |
| FDI·W | 0.484 (2.091) | 1.124 (4.204) | −0.165 (0.324) | 1.172 * (0.637) | −0.780 ** (0.392) | 1.477 ** (0.629) |
| FINLEV·W | −0.216 (0.154) | 0.083 (0.247) | −0.016 (0.024) | 0.009 (0.037) | −0.023 (0.022) | 0.011 (0.035) |
| HRCAP·W | −0.622 (0.507) | −1.486 * (0.847) | 0.047 (0.078) | 0.556 *** (0.130) | 0.094 (0.075) | 0.389 *** (0.127) |
| ED·W | −0.064 (0.300) | 0.970* (0.535) | 0.067 (0.046) | −0.100 (0.081) | 0.048 (0.043) | −0.045 (0.076 |
| Spatial rho | 0.209 *** (0.021) | 0.007 (0.033) | 0.143 *** (0.020) | 0.051 * (0.030) | 0.154 *** (0.020) | 0.068 ** (0.030) |
| N | 3124 | 3124 | 3124 | 3124 | 3080 | 3080 |
| R$^2$ | 0.930 | 0.932 | 0.809 | 0.765 | 0.822 | 0.799 |

Note: ***, **, and * are significant at the level of 1%, 5%, and 10%, respectively; the values in parentheses represent standard errors.

First of all, in the robustness test, the number of green patent applications is used to measure the level of ecological efficiency. By collecting patent application information from the State and the Intellectual Property Office, the number of green patent applica-

tions is added to the city level and treated logarithmically. From the regression results in Table 7, it is found that the economic growth target can significantly improve the ecological efficiency, with the impact effects of 4.002 and 2.783, respectively. The excessive economic growth target can significantly inhibit the ecological efficiency, and the influence effects were $-14.380$ and $-9.559$, respectively. Secondly, the SBM-DEA method is used to calculate the ecological efficiency in the robustness test, and the calculated results are used as the dependent variable for regression. Using SBM-DEA to measure ecological efficiency as a dependent variable for the robustness test, appropriate economic growth targets can significantly improve ecological efficiency, with impact effects of 0.977 and 0.933, respectively. Excessive economic growth targets lagging behind by one period significantly inhibit ecological efficiency, with impact effects of $-0.437$ and $-0.419$, respectively. Finally, through the robustness test by excluding municipalities directly under the central government, the administrative central city basically covers the most developed cities in China and has more power to create a good ecological environment and city image. At the same time, it has greater executive power and economic strength and can effectively mobilize and attract more resources for environmental protection. Therefore, administrative central cities are excluded for the robustness test. The appropriate economic growth target can significantly improve the ecological efficiency, with the impact effects of 0.856 and 0.819, respectively. The excessive economic growth target can significantly inhibit the ecological efficiency, with the impact effects of $-0.384$ and $-0.368$, respectively. Through three types of robustness tests, it is found that reasonable economic growth targets can significantly improve ecological efficiency, and the assumption that excessive economic growth targets inhibit ecological efficiency is valid, and the results are stable.

*5.5. Mechanism Test*

The fiscal expenditure structure is measured by the proportion of energy conservation and environmental protection expenditure in local fiscal expenditure (Enfis) and the proportion of fiscal science and technology expenditure in total local fiscal expenditure (Tecfis). The higher the proportion of energy conservation and environmental protection expenditure and fiscal science and technology expenditure, the higher the government's emphasis on environmental protection and scientific and technological development and the higher the ecological efficiency level. In the regression model of economic growth target to ecological efficiency, the economic growth target affects ecological efficiency through energy conservation and environmental protection expenditure and fiscal science and technology expenditure. According to the regression results in Table 8, under the economic distance matrix, the impact of environmental protection expenditure and fiscal science and technology expenditure lagging behind by one period on ecological efficiency is significantly positive, indicating that these two expenditures are path-dependent. The impact of appropriate economic growth targets on environmental protection expenditure and fiscal science and technology expenditure is significantly positive, with the impact values of 0.016 and 0.020. The impact of the above two objectives on economic growth is significantly negative, with the impact values of $-0.009$ and $-0.018$. Fiscal energy conservation and environmental protection expenditure and fiscal science and technology expenditure can significantly improve the ecological efficiency, with the impact effects of 0.087 and 0.015, respectively. Therefore, appropriate economic growth targets can increase the proportion of environmental protection and fiscal science and technology expenditures, thus improving ecological efficiency, while excessive economic growth targets inhibit the proportion of environmental protection and fiscal science and technology expenditures, thereby inhibiting the improvement of ecological efficiency.

**Table 8.** Mechanism inspection.

| | Economic Distance Matrix | | | | Geographic Adjacency Matrix | | | |
|---|---|---|---|---|---|---|---|---|
| | **Enfis** | **Techfis** | **Crste** | **Crste** | **Enfis** | **Techfis** | **Crste** | **Crste** |
| CrsteL1 | | | 0.750 *** | 0.748 *** | | | 0.734 *** | 0.730 *** |
| | | | (0.014) | (0.014) | | | (0.014) | (0.014) |
| Enfis L1 | 0.569 *** | | | | 0.550 *** | | | |
| | (0.016) | | | | (0.016) | | | |
| TechfisL1 | | 0.842 *** | | | | 0.801 *** | | |
| | | (0.015) | | | | (0.016) | | |
| L.Target | 0.016 * | 0.020 *** | 0.856 *** | 0.855 *** | 0.010 ** | 0.022 * | 0.910 *** | 0.892 *** |
| | (0.009) | (0.002) | (0.211) | (0.211) | (0.004) | (0.012) | (0.219) | (0.219) |
| L.Target$^2$ | −0.009 *** | −0.018 * | −0.386 *** | −0.385 *** | −0.005 * | −0.009 * | −0.409 *** | −0.401 *** |
| | (0.002) | (0.010) | (0.097) | (0.097) | (0.003) | (0.005) | (0.100) | (0.100) |
| Econ | 0.002 | 0.002 ** | 0.094 *** | 0.095 *** | −0.001 | 0.003 *** | 0.088 *** | 0.091 *** |
| | (0.002) | (0.001) | (0.010) | (0.010) | (0.003) | (0.001) | (0.012) | (0.012) |
| Enfis | | | 0.087 * | | | | 0.048 *** | |
| | | | (0.048) | | | | (0.009) | |
| Techfis | | | | 0.015 * | | | | 0.101 ** |
| | | | | (0.009) | | | | (0.048) |
| Stru2 | −0.005 | −0.001 | 0.034 | 0.031 | −0.004 | −0.003 | 0.016 | 0.012 |
| | (0.007) | (0.003) | (0.029) | (0.029) | (0.007) | (0.003) | (0.030) | (0.030) |
| Popden | 0.001 | 0.001 | 0.009 | 0.010 | 0.001 | 0.001 | 0.004 | 0.004 |
| | (0.002) | (0.001) | (0.007) | (0.007) | (0.002) | (0.001) | (0.007) | (0.007) |
| FDI | 0.003 | 0.051 *** | 0.351 ** | 0.349 ** | 0.006 | 0.056 *** | 0.381 ** | 0.407 *** |
| | (0.035) | (0.015) | (0.150) | (0.151) | (0.035) | (0.015) | (0.149) | (0.150) |
| FINLEV | −0.004 | 0.002 | −0.067 *** | −0.067 *** | 0.000 | 0.003 * | −0.062 *** | −0.062 *** |
| | (0.003) | (0.001) | (0.013) | (0.013) | (0.003) | (0.001) | (0.014) | (0.014) |
| HRCAP | 0.019 ** | 0.006 | 0.037 | 0.035 | 0.020 ** | 0.009 ** | 0.112 *** | 0.118 *** |
| | (0.009) | (0.004) | (0.041) | (0.041) | (0.009) | (0.004) | (0.039) | (0.039) |
| ED | 0.006 | 0.001 | 0.049 ** | 0.048 * | 0.010 * | 0.000 | 0.034 | 0.032 |
| | (0.006) | (0.003) | (0.025) | (0.025) | (0.006) | (0.003) | (0.025) | (0.025) |
| Enfis·W | | | −0.200 | | | | 0.311 ** | |
| | | | (0.188) | | | | (0.123) | |
| Tecfis·W | | | | 0.423 | | | | 0.496 ** |
| | | | | (0.342) | | | | (0.250) |
| L.Target·W | 0.128 | 0.014 | 0.653 | 0.671 | −0.170 * | −0.008 | 0.170 | 0.179 |
| | (0.131) | (0.057) | (0.557) | (0.557) | (0.096) | (0.041) | (0.408) | (0.409) |
| L.Target $^2$·W | −0.060 | −0.006 | −0.289 | −0.299 | 0.079 * | 0.005 | −0.085 | −0.088 |
| | (0.060) | (0.026) | (0.255) | (0.255) | (0.044) | (0.019) | (0.187) | (0.188) |
| Econ·W | −0.005 | −0.002 | −0.053 * | −0.060 ** | −0.002 | −0.004 ** | −0.048 *** | −0.050 *** |
| | (0.007) | (0.003) | (0.029) | (0.029) | (0.004) | (0.002) | (0.017) | (0.017) |
| Stru2·W | 0.002 | 0.006 | 0.117 | 0.130 | 0.008 | 0.005 | 0.115 ** | 0.107 ** |
| | (0.019) | (0.008) | (0.080) | (0.081) | (0.011) | (0.005) | (0.049) | (0.049) |
| Popden·W | 0.001 | −0.008 *** | 0.017 | 0.015 | −0.001 | −0.000 | 0.021 | 0.021 |
| | (0.005) | (0.002) | (0.021) | (0.021) | (0.003) | (0.001) | (0.014) | (0.014) |
| FDI·W | 0.052 | −0.101 * | 1.484 ** | 1.432 ** | 0.165 ** | 0.012 | −0.231 | −0.279 |
| | (0.139) | (0.060) | (0.594) | (0.595) | (0.070) | (0.031) | (0.302) | (0.306) |
| FINLEV·W | 0.002 | 0.001 | 0.009 | 0.008 | −0.011 ** | −0.002 | −0.016 | −0.023 |
| | (0.008) | (0.004) | (0.035) | (0.035) | (0.005) | (0.002) | (0.022) | (0.022) |
| HRCAP·W | 0.012 | 0.026 ** | 0.455 *** | 0.421 *** | 0.021 | −0.007 | 0.026 | 0.027 |
| | (0.028) | (0.012) | (0.122) | (0.126) | (0.017) | (0.007) | (0.073) | (0.073) |
| ED·W | −0.008 | 0.013 * | −0.044 | −0.051 | −0.013 | 0.005 | 0.050 | 0.051 |
| | (0.018) | (0.008) | (0.076) | (0.076) | (0.010) | (0.004) | (0.043) | (0.043) |
| Spatial rho | 0.001 | 0.068 ** | 0.060 ** | 0.059 ** | 0.141 *** | 0.116 *** | 0.162 *** | 0.157 *** |
| | (0.033) | (0.028) | (0.029) | (0.029) | (0.022) | (0.021) | (0.019) | (0.019) |
| N | 3124 | 3124 | 3124 | 3124 | 3124 | 3124 | 3124 | 3124 |
| R$^2$ | 0.448 | 0.859 | 0.802 | 0.805 | 0.417 | 0.865 | 0.837 | 0.836 |

Note: ***, **, and * are significant at the level of 1%, 5%, and 10%, respectively; the values in parentheses represent standard errors.

The empirical results obtained under the geographical adjacency matrix are similar. The impact of appropriate economic growth targets on environmental protection expenditure and fiscal science and technology expenditure is significantly positive, and the impact effects are 0.010 and 0.022. The impact of the above two objectives on economic growth is significantly negative, with the impact magnitudes of −0.005 and −0.009. Environmental protection expenditure and fiscal science and technology expenditure can significantly improve the ecological efficiency, with the impact effects of 0.048 and 0.101, respectively.

## 6. Discussion and Conclusions

### 6.1. Discussion

The purpose of this paper is to explore the relationship between economic growth targets and ecological efficiency and enrich the existing research on ecological efficiency and economic growth targets. In recent years, the Chinese government has placed great emphasis on the implementation of economic growth targets, which has greatly mobilized local governments to construct economies. However, the target management of economic growth has penetrated into areas of economic development, which has an impact on green development. There are few studies on the impact of economic growth targets and ecological efficiency.

At the level of urban ecological efficiency measurement results, we conclude that China's overall ecological efficiency level is not high, and the ecological efficiency among different regions presents a ladder distribution. Among them, the ecological efficiency in the eastern region is the highest, followed by the central region, and the western region is the lowest, which is similar to Humaira's research [65], showing a gradient decreasing pattern. The reason for this distribution is that the eastern region has rich talent, capital, policy advantages, and rich tourism resources, and the eastern cities have changed from high pollution and high energy consumption to high value-added tertiary industry, so the ecological efficiency of the eastern region is relatively high. For a long time, the natural resource endowments in the western, central, and northeastern regions have had a great impact on the regional ecological efficiency. The ecological resources are fragile, the resource development is excessive, and the environmental protection awareness is relatively backward. At the same time, they are also the main places undertaking the high-pollution and high-energy-consumption industries in the eastern region, so the ecological efficiency is relatively low. China should increase the ecological compensation for these regions [32].

In terms of the impact of economic growth target setting on ecological efficiency, this paper believes that there is an inverted U-shaped relationship between them, which is similar to the research of Chai [32]. They believe that there is a significant U-shaped relationship between economic growth targets and environmental pollution. Zhang's research supports Chai. He believes that economic growth target setting and energy conservation and emission reduction target are mutually reinforcing. Setting a higher economic growth targets will reduce environmental quality [33]. Yu et al. also believed that the reasonable setting of economic growth targets would promote the promotion of total factor productivity in prefecture-level cities and vice versa [66]. In terms of heterogeneity, the impact of economic growth target setting in the east on ecological efficiency is U-shaped, while the impact of economic growth target setting in the west and northeast on ecological efficiency is inverted U-shaped, while the relationship between the two is not significant in the central region. The existing studies are similar to the conclusions of this paper. The living standard of residents in the eastern region is relatively high, the regional industrial structure has gradually transited from heavy industry to light industry development, and the role of economic growth in economically developed regions on environmental pollution is reduced. The local government can achieve this without increasing the level of environmental pollution and resuming construction. However, the western region and the northeast region mainly rely on undertaking the industries in the eastern region, and the ecological environment is fragile. If the region is set with excessive economic growth targets, the local government will increase environmental pollution at the cost of inhibiting

the improvement of ecological efficiency [60]. Under COVID-19, the economic situation is not yet clear, and the excessively high economic growth target may "kidnap" the macro policy, which should turn the main goal of the policy into stabilizing employment and providing social security after unemployment. When China's economy is developing from high-speed to high-quality, we should make the setting of economic growth goals more beneficial to people's livelihood. In terms of impact mechanism, this paper believes that economic growth targets affect ecological efficiency through fiscal science and technology expenditure and energy conservation and environmental protection expenditure. Among them, the impact of economic growth target on fiscal science and technology expenditure and energy conservation and environmental protection expenditure is in an inverted U shape. While fiscal science and technology expenditure and energy conservation and environmental protection expenditure can significantly improve ecological efficiency, different expenditure structures have different effects on reducing environmental pollution. Although fiscal science and technology expenditure and energy conservation and environmental protection expenditure can help reduce the emission of environmental pollutants and improve environmental quality [67], the Chinese government has only decentralized expenditure rather than income. If the cities' economic growth target is set too high, local governments often adopt an extensive economic development mode in order to achieve the established economic growth target. Under the influence of the economic growth target, the competition is intensified, and they are keen to promote economic growth but ignore the green transformation of the economy [68]. They give priority to allocating fiscal resources to areas that can stimulate economic growth in the short term and promote the realization of the economic growth target at the expense of the environmental pollution [69]. Local governments are more willing to invest more funds in resource intensive projects represented by infrastructure construction to attract capital inflows, rather than attracting labor force in education [70]. Such expenditure can increase GDP in a short time, but it can bring irreversible environmental pollution and inhibit the improvement of ecological efficiency. For developing countries, optimizing the fiscal expenditure structure may be the preferred strategy. Increasing the proportion of energy conservation and environmental protection expenditure and fiscal science and technology expenditure can accelerate their transformation to being green and innovation-driven [71].

*6.2. Conclusions*

From 2007 to 2019, this paper measures ecological efficiency in 284 Chinese cities using the EBM-DEA method and examines the impact and mechanism of economic growth target setting on ecological efficiency. Based on the results of the study, the following conclusions can be drawn:

First, as for the temporal and spatial distribution characteristics of ecological efficiency, China's current ecological efficiency is still low, exhibiting an east > northeast > central > west distribution trend. The regional agglomeration situation is characterized by high–high agglomeration and low–low agglomeration.

Second, setting economic growth targets can improve ecological efficiency, while setting excessive economic growth targets can inhibit it. So, it has been proven that speeding up does not work. The high targets set for economic growth will lead to the decline of ecological efficiency. In different regions, economic growth targets have heterogeneous effects on ecological efficiency. In the eastern region, economic growth targets negatively affect ecological efficiency, and excessive economic growth targets can increase it. A reasonable economic growth target in the central, western, and northeastern regions can enhance ecological efficiency, while an excessively high economic growth target reduces it.

Third, from the perspective of the mechanism of economic growth target setting affecting ecological efficiency, a reasonable economic growth target can increase the proportion of energy conservation and environmental protection expenditure and the proportion of science and technology expenditure, thereby improving ecological efficiency. If the local government sets an excessive economic growth target, it will lead the local government to

violate the law of development, curb the proportion of energy conservation and environmental protection expenditure and science and technology expenditure, and thus inhibit the improvement of ecological efficiency.

*6.3. Implications*

According to the reasonable economic growth target in the second conclusion, the ecological efficiency should be improved, and the excessive economic growth target will inhibit the ecological efficiency. The government's target management orientation should be changed. Policymakers should coordinate the relationship between economic growth and environmental protection and guide local governments to set reasonable economic growth targets. By changing the official assessment system, the central government should weaken the proportion of economic growth in official assessment, strengthen the constraint of multi-dimensional objectives such as environmental protection and public services on local government behavior, formulate the economic growth target of the jurisdiction according to local conditions, and formulate the economic growth target according to the resource endowment and administrative level of each city. Among them, the eastern region should pay more attention to the development of high-tech industries, improve industrial technology, and accelerate the optimization and upgrading of industrial structure. The local governments in the central region, western region, and northeast region should rationally set economic growth targets to avoid irrational overweight of economic growth targets.

According to the third conclusion, the economic growth target affects ecological efficiency through environmental protection expenditure and financial science and technology expenditure, and the following policy recommendations are proposed: improve the resource optimization effect of the fiscal expenditure structure and improve the level of ecological efficiency. The scale of expenditure on energy conservation and environmental protection and fiscal expenditure on science and technology will be expanded, so that fiscal behavior will play a greater role in the adjustment of the energy structure and the transformation of the green economy. At the level of fiscal expenditure structure, we should increase the intensity of environmental protection and science and technology expenditures, ensure the smooth progress of R&D activities, enhance the enthusiasm of enterprises for innovation, and then improve the regional ecological efficiency. We will guide local governments to "compete for excellence", encourage governments at all levels to develop high-tech industries, shift funds to the cultivation of emerging technologies, and promote the transformation and upgrading of the industrial structure. We will continue to increase technical, fiscal, and experience assistance to less developed regions in the central and western regions and improve environmental quality and green technology innovation. China should increase the proportion of public opinion in governance, strengthen public supervision over environmental protection, and promote China's high-quality economic development.

**Author Contributions:** Conceptualization, C.Z. and T.L.; data curation, C.Z. and T.L.; formal analysis, C.Z.; funding acquisition, C.Z. and H.W.; methodology, C.Z. and J.L.; project administration, H.W. and X.L.; software, C.Z., J.L. and X.L.; supervision, H.W.; validation, M.X.; visualization, C.Z. and J.L.; writing—original draft, C.Z.; writing—review and editing, C.Z., J.L., T.L. and M.X. All authors have read and agreed to the published version of the manuscript.

**Funding:** This research received no external funding.

**Institutional Review Board Statement:** Ethical review and approval was not required for the study as the research does not involve humans.

**Informed Consent Statement:** Not applicable.

**Data Availability Statement:** Data available on request.

**Conflicts of Interest:** The authors declare no conflict of interest.

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
