# Peer review of "Economic Growth Target, Government Expenditure Behavior, and Cities’ Ecological Efficiency—Evidence from 284 Cities in China"

_land, doi:10.3390/land12010182_

Round 1

Reviewer 1 Report

Referee report on the manuscript “Economic Growth Target, Government Expenditure Behavior and Cities’ Ecological Efficiency – Evidence from 284 Cities in China”

I wish to thank the Editor and the authors for giving me the opportunity to read this paper. I believe it is publishable, after some major revisions.

My detailed comments follow.

I think some issues should be clarified for the readers who are not perfectly familiar with Chinese economic, political, and administrative organization. E.g., are the growth targets set by the government? Are they set at prefecture/municipal level? Is their achievement “compulsory” (i.e., what happens if they are not achieved)? I think a more thorough explanation of these aspects should be provided at the beginning of the paper, to favour the overall understanding.

The period analysed includes the 2008-2009 economic crisis. Although time fixed effects are included in the regression, this issue is not mentioned in the paper. The authors may be willing to at least discuss it.

As for the test of spatial autocorrelation, I do not think the Moran Index is exhaustive. I believe it is quite customary to have the Moran Index presented with the Lagrange Multiplier and the Robust Lagrange Multiplier (this is maybe done in Section 5.1.4?). In any case, the eventual use of spatial econometrics techniques of course reduces this problem.

I am not sure how and why an economic distance matrix should be relevant in this case. The authors may be willing to explain this point in greater depth.

I see 2 main empirical problems. The first one is related to the timing of the variables used in the econometric model. I would prefer the explanatory variables to be at least one year lagged with respect to the dependent one (the lag is used only for the dependent var). Otherwise, the timing about the decision on the economic growth target and the implementation of policy aimed at achieving it should be better explained. The second problem I see is about omitted variables. I do not think all the appropriate controls have been included in the model. In particular, the initial level of wealth (stage of development) should be included, for example in terms of GDP per capita. I believe this is extremely important. On the one hand, it would better frame the role of the economic growth target; in fact, the same target is not really the same if the starting point is different, although the target is expressed in terms of growth rates (poorer regions tend to grow faster). On the other hand, the inclusion of this control variable would potentially explain the regional differences that emerge in Table 5.

I do not think the maps displayed in Figure 3 are completely comparable, since the thresholds that identify the different categories seems to be different every time. Why is this the case? Can’t they be homogenized?

As for spatial autocorrelation in ecological efficiency, there seems to be a clear east/west divide in China. However, my impression is that this would be the case for any socio-economic variable analysed in China, since my feeling is that the country is in fact geographically quite divided between the two macro-regions.

The number of observations is not reported in the specification outputs displayed in Table 5. This should be amended, since it is a very important detail.

Minor points

The paper should be re-read since some sentences do not make sense (e.g., p. 2 “As a comprehensive indicator…”; or the “second part” missing in the presentation of the structure of the paper at the end of the introduction).

Some of the conclusions (especially in terms of what the government should/shouldn’t do) do not descend directly from the empirical results.

Author Response

Reviewer1

  1. COMMENT:I think some issues should be clarified for the readers who are not perfectly familiar with Chinese economic, political, and administrative organization. E.g., are the growth targets set by the government? Are they set at prefecture/municipal level? Is their achievement “compulsory” (i.e., what happens if they are not achieved)? I think a more thorough explanation of these aspects should be provided at the beginning of the paper, to favour the overall understanding.

Answer: Thank you for your careful review. According to your suggestion, we have added the following background to Introduction. See the text for details.Although the government performance assessment is increasingly diversified, the economic growth is the easiest to measure, so it is still the most important assessment indicator [6]. From the 12th National People's Congress of the CPC to the 18th National People's Congress, the goal of doubling economic growth has been clearly put forward, and the goal of economic growth has become the performance standard publicly committed by governments at all levels [7]. In the process of transforming from a planned economy to a market economy, China has gradually formed a vertical target management system along the path of socialism with characteristics [8]. Under China's political pyramid, the central government holds the power of personnel, and local governments have the ability to intervene in economic development. Officials with high pressure on political promotion will interfere with the economy and operation of enterprises. The central government leads China's economic growth through economic growth goals, which are broken down to governments at all levels through administrative levels. The economic growth target is not only the assessment of the superior government to the subordinate government, but also the commitment of the subordinate government to the superior government. China's fiscal decentralization system gives officials full control over resources and administrative decision-making power, enabling local officials to have a strong ability to intervene and dominate economic development [9]. Under this system background, if local officials want to achieve performance appraisal standards and career promotion, they must use various resources to promote economic growth”.

  1. COMMENT:The period analysed includes the 2008-2009 economic crisis. Although time fixed effects are included in the regression, this issue is not mentioned in the paper. The authors may be willing to at least discuss it.

Answer: Thank you for your careful review. In this paper, the data of 2008-2009 are empirically tested with the two-way fixed space Durbin model, and the following regression results are obtained, which are consistent with the benchmark regression results of this paper.

Table1 Regression results

Geographic adjacency matrix

Economic adjacency matrix

Inverse distance matrix

Main

Main

Main

L.Target

0.883***

(0.257)

0.829***

(0.246)

0.961***

(0.253)

L.Target 2

-3.874***

(0.824)

-3.582***

(0.788)

-4.035***

(0.817)

Econ

0.272***

(0.021)

0.267***

(0.020)

0.274***

(0.021)

Stru2

-0.002

(0.054)

0.020

(0.050)

0.021

(0.054)

Popden

0.001

(0.013)

0.004

(0.013)

-0.001

(0.013)

FDI

0.440

(1.042)

1.072

(1.075)

0.133

(1.051)

FINLEV

-0.037

(0.038)

-0.024

(0.037)

-0.022

(0.038)

HRCAP

0.228

(0.141)

0.095

(0.150)

0.337**

(0.145)

ED

-0.076

(0.076)

-0.049

(0.075)

-0.050

(0.076)

L.Target·W

1.270**

(0.542)

-0.004

(0.558)

5.104**

(2.245)

L.Target2 ·W

-3.219**

(1.508)

1.250

(1.765)

-12.714*

(6.942)

Econ·W

0.004

(0.039)

0.005

(0.060)

-0.121

(0.109)

Stru2·W

0.012

(0.087)

-0.018

(0.175)

-0.086

(0.187)

Popden·W

-0.027

(0.029)

-0.015

(0.034)

0.008

(0.084)

FDI·W

0.441

(2.288)

-0.915

(2.392)

-1.705

(5.310)

FINLEV·W

0.036

(0.067)

0.040

(0.111)

0.059

(0.152)

HRCAP·W

0.036

(0.263)

1.259***

(0.390)

1.897**

(0.865)

ED·W

0.129

(0.142)

0.026

(0.222)

0.405

(0.344)

Spatial rho

0.085*

(0.059)

0.031***

(0.004)

0.103*

(0.069)

N

568

568

568

R2

0.179

0.214

0.168

  1. COMMENT:As for the test of spatial autocorrelation, I do not think the Moran Index is exhaustive. I believe it is quite customary to have the Moran Index presented with the Lagrange Multiplier and the Robust Lagrange Multiplier (this is maybe done in Section 5.1.4?). In any case, the eventual use of spatial econometrics techniques of course reduces this problem.

Answer: Thank you for your careful review. We have added LM test to provide more empirical evidence for the selection of spatial econometric models. See the table2 for details

Table 2 LM Test

Mixed regression

Fixed regions

Fixed time

Fixed regions and time

LM test no spatial lag, probability

230.4370

(0.000)

427.5657

(0.000)

219.9619

(0.000)

413.7267

(0.000)

robust LM test no spatial lag, probability

2.1882

(0.139)

22.1786

(0.000)

5.3913

(0.020)

25.9987

(0.000)

LM test no spatial error, probability

287.2216

(0.000)

431.0128

(0.000)

252.5502

(0.000)

402.5312

(0.000)

robust LM test no spatial error, probability

58.9729

(0.000)

25.6257

(0.000)

37.9796

(0.000)

14.8031

(0.000)

  1. COMMENT:I am not sure how and why an economic distance matrix should be relevant in this case. The authors may be willing to explain this point in greater depth.

Answer: Thank you for your careful review. In the original text, we added the explanation under the economic distance matrix in the benchmark regression. See the following for details: The spatial spillover effect of regional ecological efficiency is not only characterized by geographical distance, but also affected by the gap of economic development level. If only geographical distance is used to measure the spatial spillover effect of ecological efficiency, there will inevitably be deviation. Therefore, this paper selects the per capita GDP of prefecture level cities as the matrix element, constructs the economic distance matrix and carries out spatial regression. The autocorrelation coefficient under the economic distance matrix is positive, and significant at the 1% and 5% levels, indicating that the ecological efficiency of a city will be affected by cities with closer economic attributes. The ecological efficiency lagging behind the first stage has a significant impact on the local ecological efficiency, which is 0.750. The reasonable economic target setting lagging behind by one period can effectively improve the local ecological efficiency. Under the two-way fixed effect and dynamic two-way fixed effect models, the size is 0.451 and 0.859 respectively. A reasonable economic target setting can significantly improve the local ecological efficiency, with the effect size of 0.115. Cities with similar economic development levels have an imitation effect in setting economic growth goals. The reasonable economic target setting lagging behind by one period inhibited the improvement of local ecological efficiency, with the effect of -0.039, but not significant. In the dynamic spatial Dubin model, neighboring regions with similar economic development levels lag behind the reasonable economic goal setting of Phase I to promote the improvement of local ecological efficiency, but the effect is not significant. Neighbouring regions with similar economic development levels lag behind the excessive economic goal setting of Phase I to significantly inhibit the improvement of local ecological efficiency, the size is -0.295.

  1. COMMENT:I see 2 main empirical problems. The first one is related to the timing of the variables used in the econometric model. I would prefer the explanatory variables to be at least one year lagged with respect to the dependent one (the lag is used only for the dependent var). Otherwise, the timing about the decision on the economic growth target and the implementation of policy aimed at achieving it should be better explained. The second problem I see is about omitted variables. I do not think all the appropriate controls have been included in the model. In particular, the initial level of wealth (stage of development) should be included, for example in terms of GDP per capita. I believe this is extremely important. On the one hand, it would better frame the role of the economic growth target; in fact, the same target is not really the same if the starting point is different, although the target is expressed in terms of growth rates (poorer regions tend to grow faster). On the other hand, the inclusion of this control variable would potentially explain the regional differences that emerge in Table 5.

Answer: Thank you for your careful review. We have delayed the setting of economic growth targets by one period. See the text for details. In addition, we also accepted the reviewer's suggestion to regress the economic growth target as a control variable. See the text for details.

  1. COMMENT:I do not think the maps displayed in Figure 3 are completely comparable, since the thresholds that identify the different categories seems to be different every time. Why is this the case? Can’t they be homogenized?

Answer: Thank you for your careful review. We did not mark specific values in the map, which led to doubts among reviewers. Considering that the natural breakpoint method in ArcGIS cannot compare the ecological efficiency of different years, this study uses the natural segment point method and the bisection method in ArcGIS to grade the ecological efficiency, ensuring the comparability of the ecological efficiency in 2007-2019. The specific classification is as follows: strongly ineffective (0, 0.4], ineffective (0.4, 0.6], weakly ineffective (0.6, 0.8], weakly effective (0.8, 1.0], effective (1.0,1.2].

  1. COMMENT:As for spatial autocorrelation in ecological efficiency, there seems to be a clear east/west divide in China. However, my impression is that this would be the case for any socio-economic variable analysed in China, since my feeling is that the country is in fact geographically quite divided between the two macro-regions.

Answer: Thank you for your careful review. Since the East, Middle, West and Northeast are at different stages of development, and the formulation of economic growth goals is different, there must be greater differences in the government's behavior choices. Therefore, there may be regional differences in the impact of economic growth goals on ecological efficiency. We want to compare the differential impact of economic growth objectives in the east, middle, west and northeast on ecological efficiency. See the text table for specific differences

  1. COMMENT:The number of observations is not reported in the specification outputs displayed in Table 6. This should be amended, since it is a very important detail.

Answer: Thank you for your careful review. Due to our negligence, the number of observations in Table 6 is missing, and the number of cities has been supplemented.

  1. COMMENT:The paper should be re-read since some sentences do not make sense (e.g., p. 2 “As a comprehensive indicator...”; or the “second part” missing in the presentation of the structure of the paper at the end of the introduction).

Answer: Thank you for your careful review. We have supplemented the part of "comprehensive indicators..." on page 2. As a comprehensive indicator, ecological efficiency can measure the level of economic development and the coordination level of environmental quality. We add the missing "Part two" at the end of the introduction.

  1. COMMENT:Some of the conclusions (especially in terms of what the government should/shouldn’t do) do not descend directly from the empirical results.

Answer: Thank you for your careful review. Based on your suggestion, we have added the conclusion basis before the proposal is put forward.

Reviewer 2 Report

Dear authors,

This theme is a very interesting because in ecological point of view the whole Earth is in the same problem. Problem is even more accelerated when we observe the situation from our own backyard (our own country). As we are all interconnected you have to extent your exploration within the situation all over the world. In that sense I know that European Green Plan has been proposed and adopted in order to mitigate ecological problems. You should explore that example and compare it with Chinese one as well as explore worldwide situation.

You need to consider also a period of worldwide crises in previous decade, as well as give possible implications in recent three years due to the Covid and Ukraine war economic possible implication (eventhough it is not the period investigated - it would be good to at least refer to it). And authors have not taken into account (or i m not aware of it) a level of development (GDP per capita).
At the same time exploration of previous investigations is based only on Chinese examples which degrade the relevance of the work. If they spread it in more wider context its significance would be very high. In connection to this conclusions needs to be upgraded. 

Kind regards

Author Response

Reviewer2

  1. COMMENT:This theme is a very interesting because in ecological point of view the whole Earth is in the same problem. Problem is even more accelerated when we observe the situation from our own backyard (our own country). As we are all interconnected you have to extent your exploration within the situation all over the world. In that sense I know that European Green Plan has been proposed and adopted in order to mitigate ecological problems. You should explore that example and compare it with Chinese one as well as explore worldwide situation.

Answer: Thank you for your careful review. Comparative research among different countries is our future work direction. In the future, we will compare and explore the relationship between economic growth goal setting and ecological efficiency in China and other countries through case study methods and empirical research methods.

  1. COMMENT:You need to consider also a period of worldwide crises in previous decade, as well as give possible implications in recent three years due to the Covid and Ukraine war economic possible implication (eventhough it is not the period investigated - it would be good to at least refer to it). And authors have not taken into account (or not aware of it) a level of development (GDP per capita).At the same time exploration of previous investigations is based only on Chinese examples which degrade the relevance of the work. If they spread it in more wider context its significance would be very high. In connection to this conclusions needs to be upgraded.

Answer: Thank you for your careful review. In the discussion part, this paper supplements the existing problems and setting direction of China's economic growth goals since COVID-19. See text P33 for details

“Under the COVID-19, the economic situation is not yet clear, and the excessively high economic growth target may "kidnap" the macro policy and lead to flooding, which can turn the main goal of the policy into stabilizing employment and providing social security after unemployment. At the moment when China's economy is developing from high-speed to high-quality, we should make the setting of economic growth goals closer to and more beneficial to people's livelihood”

Reviewer 3 Report

Dear authors,

I would like to congratulate you for the proposal launched for this magazine. I believe it is an innovative, enriching and suitable work for publication.

Best regards.

Author Response

Thank you for your comments.